# Global Balanced Wind Derived from SABER Temperature and Pressure Observations and its Validations

Xiao Liu[1,2], Jiyao Xu[2,3]*, Jia Yue[4#] , You Yu[5], Paulo P. Batista[6], Vania F. Andrioli[2,6], Zhengkuan Liu[2], Tao Yuan[7], Chi Wang[2], Ziming Zou[2], Guozhu Li[5], James M. Russell III[4]

[1]Henan Engineering Laboratory for Big Data Statistical Analysis and Optimal Control, School of Mathematics and Information Sciences, Henan Normal University, Xinxiang, 453000, China
[2]State Key Laboratory of Space Weather, Center for Space Science and Applied Research, Chinese Academy of Sciences, Beijing, 100190, China
[3]School of Astronomy and Space Science, University of the Chinese Academy of Science, Beijing, 100049, China
[4]Atmospheric and Planetary Sciences, Hampton University, Hampton, VA 23668, USA
[5]Key Laboratory of Ionospheric Environment, Institute of Geology and Geophysics, Chinese Academy of Sciences, Beijing, 100029, China
[6]Heliophysics, Planetary Science and Aeronomy Division, National Institute for Space Research (INPE), Sao Jose dos Campos, Sao Paulo, Brazil
[7]Center for Atmospheric and Space Sciences, Utah State University, Logan, UT, 84322, USA

*Correspondence to*: Jiyao Xu (xujy@nssc.ac.cn)

**Abstract.** Zonal winds in the stratosphere and mesosphere play important roles in the atmospheric dynamics and aeronomy. However, the direct measurement of winds in this height range is difficult. We present a dataset of the monthly mean zonal wind in the height range of 18-100 km and at latitudes of 50°S-50°N from 2002 to 2019, which is derived by the gradient balance wind theory and the temperature and pressure observed by the SABER instrument. The tide alias above 80 km at the equator is replaced by the monthly mean zonal wind measured by a meteor radar at 0.2°S. The dataset (named as BU) is validated by comparing with the zonal wind from MERRA2 (MerU), UARP (UraU), HWM14 empirical model (HwmU), meteor radar (MetU) and lidar (LidU) at seven stations from around 50°N to 29.7°S. At 18-70 km, BU and MerU have (1) nearly identical zero wind lines, (2) year-to-year variations of the eastward/westward wind jets at middle and high latitudes, (3) the quasi-biennial oscillation (QBO) and semi-annual oscillation (SAO), especially the disrupted QBO in early 2016. The comparisons among BU, UraU and HwmU show good agreement in general below 80 km. Above 80 km, the agreements among BU, UraU, HwmU, MetU and LidU are good in general, except some discrepancies at limited heights and months. The BU data are archived as netCDF files and can be available at https://dx.doi.org/10.12176/01.99.00574 (Liu et al., 2021). The advantages of the global BU dataset are the large vertical extent (from the stratosphere to the lower thermosphere) and long-term duration (2002-2019). The BU data is useful to study the temporal variations with periods ranging from seasons to decades at 50°S-50°N. It can also be used as the background wind for atmospheric wave propagation.

---

# Now at Catholic University of America, Washington, DC 20064, USA

# 1 Introduction

Zonal mean zonal wind in the middle and upper atmosphere is critical to the propagation and dissipation (filters or prohibits) of the atmospheric waves (e.g., gravity waves, tides, and planetary waves) while the waves propagate against or along the zonal winds (Forbes, 1995; Fritts and Alexander, 2003). On the other hand, the waves dissipate their energy and momentum in the mean flow and accelerate or decelerate the mean wind. This changes the atmospheric thermal and wind structures and even reverse the wind directions (McLandress, 1998; Zhang and Shepherd, 2005; Watanabe and Miyahara, 2009; Liu et al., 2014; Becker and Vadas, 2018). Thus, the zonal wind climatology and the mean wind-waves interactions are important topics in atmosphere dynamics.

Ground-based observations (e.g., radiosondes, rockets, Fabry-Perot interferometers, radars and lidars, etc.) have a long history of wind measurements. They provide horizontal winds in a limited altitude range and/or locations, but have a wide and even full local time (LT) coverage. On the other hand, satellite observations provide global observations of winds in the middle and upper atmosphere in a limited LT or height ranges. For example, the High Resolution Doppler Imager (HRDI) on the Upper Atmosphere Research Satellite (UARS) observed the winds in 10-40 km and 65-110 km of during day time from 1991 to 1999 (Hays et al., 1993). The Wind Imaging Interferometer (WINDII) on the UARS observed the winds in 90-120 km during both day and night times from 1991 to 1997 (Zhang and Shepherd, 2005; Shepherd et al., 2012). Onboard the Thermosphere, Ionosphere, Mesosphere Energetics and Dynamics (TIMED) satellite, the TIMED Doppler Interferometer (TIDI) observed wind in 70-115 km during day and 80-105 km during night since 2002 (Killeen et al., 2006). The TIDI wind data are mainly used to study tides and planetary waves due to the uncertainties in its absolute zero wind (Niciejewski et al. 2006; Wu et al., 2006, 2008; Gu et al., 2013). On September 10th, 2019, NASA's Ionospheric Connection Explorer (ICON) was launched to study the earth's ionosphere (Immel et al., 2018). The Michelson Interferometer for Global High-Resolution Thermospheric Imaging (MIGHTI) onboard the ICON satellite has two identical sensor units: MIGHTI-A and MIGHTI-B, which can be used to retrieve temperature at 90-115 km, the line-of-sight winds and the vector winds at 90-300 km (Englert et al., 2017; Harding et al., 2017; Stevens et al., 2018). These ground-based and satellite observations, as well as rocket soundings are useful to construct empirical wind models, such as the COSPAR International Reference Atmosphere (CIRA-86) (Fleming et al., 1990), the Horizontal Wind Models (HWM) (Drob et al., 2008, 2015; Emmert et al., 2008) and the Upper Atmosphere Research Satellite (UARS) Reference Atmosphere Project (URAP) wind climatology (Swinbank and Ortland, 2003).

These observations are useful to constrain the reanalysis wind data, such as the Modern-Era Retrospective analysis for Research and Applications, version 2 (MERRA2) (Molod et al., 2015; Gelaro, et al., 2017) and the European Centre for Medium-Range Weather Forecasts ERA5 (Hoffmann, et al., 2019; Hersbach et al., 2020). A recent study by Ern et al. (2021) showed that both MERRA2 and ERA5 capture the semi-annual oscillations (SAO) in the stratopause and the lower mesosphere at around the equator. In the middle mesosphere, MERRA2 produces a reasonable SAO due to the assimilated Aura Microwave Limb Sounder (MLS) data (Schwartz et al., 2008; Molod et al., 2015). Above 70 km, the mesopause SAO produced by ERA5

is stronger than that by MERRA2. This is because the stronger damping of MERRA2 reduces the amplitude of the mesopause

SAO. These observations and models greatly improved our knowledge on the winds in the middle and upper atmosphere.

The zonal winds provided by CIRA-86 and UARP are in the sense of global climatology and at 0-120 km and 0-110 km, respectively (Fleming et al., 1990, Swinbank and Ortland, 2003). The zonal and meridional winds provided by HWM series are a function of day of the year and LT and from surface to thermosphere (~500 km) (Drob et al., 2008, 2015; Emmert et al., 2008). The zonal winds from these models are useful to study the seasonal variations but not for the variations with periods

longer than one year. The zonal winds provided by reanalysis data (e.g., MERRA2 and ERA5, etc.) are in the height range from surface to ~70-80 km and are useful to study the variations with periods from several days to several years (Gelaro et al., 2017; Hoffmann et al., 2019; Ern et al., 2021).

In the current state, the direct global measurement of zonal wind in the upper stratosphere and mesosphere is difficult, and the model-inherent damping in the upper model levels of MERRA2 and ERA5 is still a challenge to get realistic wind in the

mesosphere and lower thermosphere (MLT) region (Ern et al., 2021). A candidate is combining the observations of temperature and pressure with balance wind theory to get zonal wind in the MLT region. Smith et al. (2017) have derived the balance wind (BU) at around the equator, which is based on the gradient wind balance theory (Randel, 1987) and observation data. The observation data are: (1) the geopotential height observed by the MLS (Schwartz et al., 2008) from 2004 to 2016, and (2) the pressure and temperature measured by the Sounding of the Atmosphere Using Broadband Emission Radiometry (SABER)

instrument (Russell et al., 1999) on the TIMED satellite from 2002 to 2015. They showed the SAO of zonal wind and their relations with quasi-biennial oscillations (QBO) in the tropical upper stratosphere and mesosphere. Smith et al. (2017) noted that the BU is reasonable below ~84 km but not above ~84 km. This is because the aliasing of diurnal tide to mean wind is notable above 84 km (Mclandress et al., 2006; Xu et al., 2009). This is because the diurnal tide is prominent and exhibits short-term (one to serval days) variations. The full diurnal cycle is composed by the data from many days (e.g., 60 days for SABER

observations). Thus, the obtained are the mean of diurnal tides over these days. However, the short-term variations of diurnal tides are still in the background and alias the derived winds based on the gradient wind balance theory.

The focus of this work is to provide a global dataset of the monthly mean zonal wind (short for BU dataset) at 18-100 km, which is based on the gradient wind balance theory and the temperature and pressure measured by the SABER instrument. The BU dataset extends from 2002 to 2019 and from 50°S to 50°N. To overcome the unrealistic BU above 84 km over the

equator (Smith et al., 2017), we replace the BU above 80 km by the zonal wind measured by the meteor radar at Kototabang (0.20°S, 100.32°E). The validation of the BU dataset will be performed by comparing with those from MERRA2, UARP, and meteor radar and lidar observations from around 50°N to 29.7°S.

The advantages of the global BU dataset are their large vertical extent and long-term temporal coverage. The vertical extent is from the stratosphere to the lower thermosphere. The temporal coverage is from 2002 to 2019. Thus, the BU dataset can be

used to study the global variations of zonal wind in time scales ranging from season to decades and from the stratosphere to the lower thermosphere. These variations include SAO, AO, QBO and ENSO (El Niño–Southern Oscillation, periods of 2-8 years, Baldwin and O'Sullivan, 1995). Although QBO and ENSO are originated from the lower atmosphere or sea surface,

their influences are global and can extend to the stratosphere or even higher heights and latitudes (Baldwin and O'Sullivan, 1995; Baldwin et al., 2001). Moreover, the interactions among SAO, AO, QBO and ENSO are also important in modulating global atmospheric waves and composition from the stratosphere to the lower thermosphere (e.g., Xu et al., 2009; Liu et al., 2017; Diallo et al., 2018; Ern et al., 2011, 2014, 2021; Kawatani et al., 2020).

## 2 Data and Method

### 2.1 Data Description

MERRA2 is the new version of atmospheric reanalysis dataset developed by NASA's Global Modeling and Assimilation Office (Molod et al., 2015; Gelaro, et al., 2017). We use the 72 levels (~0-75 km) assimilated meteorological fields, which have time, longitude and latitude intervals of 3 hours, 0.625° and 0.5°, respectively. The MERRA2 zonal winds are interpolated to uniform vertical grids from 2 km to 72 km with a step of 1 km. Then they are averaged in a latitude band of 5° with an overlap of 2.5° in each month. Such that the monthly zonal mean (MerU) wind can be obtained and will be used to validate the BU at 50°N-50°S.

The URAP zonal wind (UraU) is based on the winds observed by the HRDI instrument, the stratospheric assimilation data from the Met Office, and the gradient wind balance calculated from URAP temperature data from April 1992 to March 1993 (Swinbank and Ortland, 2003). The UraU mainly represents the period of 1992-1993 and can be used as a reference wind dataset in the climatology sense. The UARP zonal wind (UarU) covers from 1000 hPa (~z=0 km) to $4.6 \times 10^{-5}$ hPa (~z=110 km) and from 80°S to 80°N with equally interval of 4° (Swinbank and Ortland, 2003).

HWM14 is the latest version of HWM, which provides the global zonal and meridional winds from the surface to thermosphere (~500 km) and their variations with LT and day of year (Drob et al., 2015). After setting the longitude to be 0 and changing the LT from 0 to 23 at each day, we get the hourly zonal winds from 18 to 100 km with a step of 1 km and from 50°S to 50°N with a latitude interval of 2.5°. Then the daily mean zonal wind is calculated by average the hourly zonal winds in one day. Finally, we change the day numbers from 1 to 365 to get the monthly mean zonal winds, which is referred to be the HWM14 zonal wind (HwmU). We note that the monthly mean zonal wind does not depend on longitude since the stationary planetary waves and migrating tides reproduced by HWM14 can be removed on a time scale of one month (Drob et al., 2015).

The zonal winds measured by meteor radars and lidar are used to improve the BU over the equator and to validate BU at middle and high latitudes. The radars' locations and their data periods are listed in Table 1. For the meteor radars, they measure the zonal and meridional winds at 80-100 km with a vertical interval of 2 km and a temporal interval of 1 hour. The zonal winds measured by these meteor radars are averaged over each calendar month to get the monthly zonal winds (MetU). The MetU spans from 53.5°N to 29.7°S and is useful to compare the BU at 80-100 km and in both the northern (NH) and southern hemispheres (SH). Especially, the MetU at KT (0.20°S) can be used to replace the tidal aliased BU over the equator since the aliasing for tides on BU (Mclandress et al., 2006; Xu et al., 2009; Smith et al., 2017). Affected by the weather conditions, the zonal winds measured by the CSU lidar (LidU) from 2002 to 2008 are rearranged in a composite year according to calendar

month in 80-100 km with a vertical interval of 0.5 km. The LidU is used to compare with the BU in the climatology sense. The detailed description of the meteor radars and lidar, as well as their measurement uncertainties, can be found in the references listed in Table 1.

The BU is derived from the temperature and pressure profiles (level 2A, version 2.07) measured by the SABER instrument (Russell et al., 1999) from 2002 to 2019. These profiles cover ~15-110 km and latitudes of 53°S-83°N or 83°S-53°N. The temperature accuracy is 1-3 K from 30 to 80 km and 5-10 K from 90 to 100 km (Remsberg et al., 2008). The detailed procedure of deriving BU is described in the next subsection.

**2.2 Method of Deriving Balanced Wind**

The method of deriving BU is ascribed to the following two steps. The first step is to derive the zonal mean temperature and pressure. All the original profiles measured by the SABER instrument are interpolated linearly to 18-108 km with vertical interval of 1 km. Then these profiles are sorted into latitude bands, which have width of 5° with an overlap of 2.5° and extend from 50°S to 50°N. At each latitude band ($\varphi$) and height ($z$), the temperature can be expressed as $T(t_{UT}, \lambda)$ ($\lambda$ is longitude). Then the zonal mean temperature in each universal time (UT) day, $T_{ZMUT}(t_{ZMLT})$, for the ascending and descending nodes, respectively, can be expressed as,

$$T_{ZMUT}(t_{ZMLT}, z, \varphi) = \frac{1}{2\pi}\int_0^{2\pi} T(t_{UT}, \lambda, z, \varphi)d\lambda, \ t_{ZMLT}(z, \varphi) = \frac{1}{2\pi}\int_0^{2\pi} t_{LT}(\lambda, z, \varphi)d\lambda. \tag{1}$$

Thus, $T_{ZMUT}(t_{ZMLT}, z, \varphi)$ excludes the nonmigrating tides and stationary planetary waves but contains the zonal mean temperature in a LT day ($T_{ZMLT}(t_{ZMLT}, z, \varphi)$, short for $\bar{T}(z, \varphi)$) and migrating tides (Xu et al., 2007; 2014, Gan et al., 2014). It takes about 60 days to get a nearly complete LT coverage for the SABER measurements. Thus, $T_{ZMLT}(t_{ZMLT}, z, \varphi)$ and migrating tides can be calculated by the least square fitting $T_{ZMUT}(t_{ZMLT}, z, \varphi)$ in a time window of 60-day and forward 1 day. Here both the ascending and descending data are used for the fitting. The fitting function is expressed as,

$$T_{ZMUT}(t_{ZMLT}, z, \varphi) = \bar{T}(z, \varphi) + A_n \cos[n\omega_0(t_{ZMLT} - t_n)]. \tag{2}$$

Here, $\omega_0 = 2\pi/(24 \text{ hour})$, $n = 1, 2, 3, 4$ is the frequency (unit of 1/day) of migrating tides. $A_n$ and $t_n$ are the amplitude and phase of the migrating tides with frequency of $n$. $\bar{T}(z, \varphi)$ is the zonal mean temperature in a LT day. In the same way, the zonal mean pressure $\bar{p}(z, \varphi)$ can be obtained.

The second step is to calculate the BU from $\bar{T}$ and $\bar{p}$. The zonal mean of the momentum equation in the zonal direction is used to calculate the gradient balance wind (Randel, 1987; Fleming et al., 1990; Xu et al., 2009),

$$\frac{\bar{u}^2}{a}\tan\varphi + f\bar{u} = -\frac{1}{a\bar{\rho}}\frac{\partial\bar{p}}{\partial\varphi}. \tag{3}$$

Here, $f = 2\Omega\sin\varphi$ is the Coriolis factor, $\Omega = 2\pi/(24 \times 60 \times 60)$ is the earth rotation frequency (unit of rad·s⁻¹), $a$ is the radius of the earth. $\bar{u}$ and $\bar{\rho} = \bar{p}/R\bar{T}$ are the BU and zonal mean density, respectively. $R$ is the gas constant for dry air. Eq. (3) has been successfully applied to the latitude bands of 70°S-8°S and 8°N-70°N to get zonal mean wind (Fleming et al., 1990; Smith et al., 2017). We restrict Eq. (3) at 10°N-50°N and 10°S-50°S due to the un-continuous sampling of the SABER measurements poleward of 53°N/S. At around the equator, the solution of Eq. (3) is an indeterminate form of 0/0 as $\varphi \to 0$

and can be solved through the L'Hôpital's rule if we get continuous values of $\bar{p}$ and $\bar{\rho}$. In fact, only the discrete values $\bar{p}$ and $\bar{\rho}$ with latitude interval of 2.5° can be obtained from observations. To apply Eq. (3) at the equator, one need to differentiate Eq. (3) with $\varphi$. As $\varphi \to 0$, we have $\tan \varphi \to \varphi$, $\sin \varphi \to \varphi$. Thus, Eq. (3) can be simplified as (Fleming et al., 1990; Swinbank & Ortland, 2003),

$$\bar{u} = -\frac{1}{2\Omega a \bar{\rho}} \frac{\partial^2 \bar{p}}{\partial \varphi^2}. \qquad (4)$$

Here the BU below 80 km is obtained from Eq. (4). Due to the alias of diurnal tide to the BU above 84 km at the equator (Mclandress et al., 2006; Xu et al., 2009; Smith et al., 2017), the BU in 80-100 km calculated by Eq. (4) will be replaced by the MetU at KT (0.20°S). Consequently, the reconstructed BU should be reliable throughout the height ranges from 18 to 100 km. The replaced BU will be described in the next subsection.

At 2.5°N-7.5°N and 2.5°S-7.5°S, the BU is estimated by a cubic spline interpolation of the BU at 10°N-50°N, 10°S-50°S and the reconstructed BU at the equator (Smith et al., 2017).

**2.3 Modification of Equatorial Balance Wind by the Wind Measured by Meteor Radar at Koto Tabang**

The MetU measured at KT (0.20°S) provides a unique advantage to modify the BU at the equator. Such that one can get reliable BU up to 100 km. Fig. 1 shows the daily mean (black) and monthly mean (red) zonal wind at 86 km measured by the meteor radar at KT station. We can see that the wind data are continuous from November 2002 to September 2017 except during some months in 2013 and 2014. To match the SABER measurements from 2002 to 2019, we have to get a continuous dataset from 2002 to 2019 through filling the missing data in 2013 and 2014, and extending the data backward to February 2002 and forward to December 2019.

The continuous dataset is constructed by multiple linear regression (MLR, Chapter 6 of Kutner et al. (2005)) through the following three steps: (1) separating the data into 4 segments (a: 2002-2007, b: 2006-2011, c: 2010-2015, d: 2014-2019) with an overlap of 24 months. The separation seems arbitrary but includes continuous observation in segment b, which is regarded as a reference segment and used as a predictor variable in MLR. Moreover, the separation retains as many as more observations in the other three segments to improve the confidence level of MLR. The overlap of 24 months is to cover the missing observation from October 2017 to December 2019. (2) Constructing the predictor variables of MLR. The first predictor variable is the constant of 1. Its regression coefficient represents the mean wind of each segment. The second predictor variable is the wind data in segment b. Fig. 1 shows that the temporal variations of winds in segments a, c, and d are similar to that in the segment b but have slightly different oscillations' amplitudes. After inspecting each segment, we see that the prominent oscillations have periods of 12 and 6 months, which are used as the 3rd and 4th predictor variables. To reproduce more realistic regression, we also include oscillations of periods of 36, 24, 4, and 3 months as the 5th-8th predictor variables. The predictor variables can be summarized as a constant of 1, the wind data in segment b, oscillations with periods of 36, 24, 12, 6, 4, and 3 months. (3) Using the predictor variables mentioned in step (2), we perform MLR on the segments a, c and d, respectively. Then the missing observations are filled with MLR predictions (shown as blue line with dots in Fig. 1).

To quantify the rationality of the MLR method, we used $R^2$ score, which is the ratio of the variations in the observation data explained by the model and defined as,

$$R^2 = 1 - \sum_{i=1}^{n}(y_i - f_i)^2 / \sum_{i=1}^{n}(y_i - \bar{y})^2, \bar{y} = \frac{1}{n}\sum_{i=1}^{n} y_i. \tag{5}$$

Here, $y_i$ and $f_i$ are the observation data and model results with data numbers of $n$, respectively. The best $R^2$ score is 1 when the predicted values are the same as the observation data. For segments a, c, and d, their $R^2$ scores are 0.63, 0.59, and 0.65, respectively. And their available observation months are 60, 57, and 34, respectively. It should be noted that the $R^2$ scores increase with the increasing number of predictor variables. However, the increasing number of predictor variables reduces the robustness of the model when the available observation months are short (e.g., segment c). Thus, the predictor variables chosen here are an optimal compromise between the $R^2$ score and the robustness of the MLR model. Fig. 1 shows that the MLR fittings coincide well with the observed monthly mean zonal winds when observations were available. It is reasonable to expect that the MLR predictions in the time intervals of missing observations are reliable (e.g., 2013 and 2014, before November 2002 and after September 2017) and can be used to construct BU.

After applying the MLR on the zonal wind measured by the KT meteor radar at 80-100 km, we obtain a continuous dataset of MetU. This continuous dataset is composed by the observed data when they were available and predicted values when the observed data were missing. Then this continuous dataset is used to replace the BU calculated by Equation (4) at 80-100 km. Combined with the BU calculated by Eq. (4) at 18-80 km and the MetU measured by the KT meteor radar at 80-100 km, we can get a reconstructed BU at 18-100 km. Fig. 2 shows the BU at 18-100 km calculated by Eq. (4) at the equator (a) and the reconstructed BU (b and c). We note that the reconstructed BU is smoothed by 3-point running mean in height and time, respectively. Fig. 2 shows that the BU above 80 km (a) is in the eastward direction during most of months, which is opposite to the replaced BU (b and c). This is because the BU above 80 km shown in Fig. 2(a) is aliased by the diurnal tide over the equator (Mclandress et al., 2006; Xu et al., 2009; Smith et al., 2017).

## 3 Validations of the Balance Wind

To validate the BU derived from the SABER observations and modified by meteor radar wind observations near the equator, we will compare the BU with (1) monthly mean zonal winds from MERRA2 data (MerU), (2) the UARP wind (UarU) and the zonal wind calculated from HWM14, (3) the zonal winds observed by meteor radars (MetU) at latitudes of 29.7°S-53.5°N and a Na lidar (LidU) at 40.6°N.

## 3.1 Comparisons with the Wind from MERRA2

First of all, we should note that the BU data are derived from the temperature and pressure profiles measured by the SABER instrument and the zonal wind observed by a meteor at Koto Tabang (0.20°S). None of these data are assimilated in assimilated in MERRA2. Thus, BU and MerU are independent. Figs. 2b and 2c, and Figs. 3-4 compare the BU derived from SABER observations and the monthly mean zonal winds from MERRA2 data (MerU) at the equator, the middle (30°N/S) and high

(50°N/S) latitudes. To perform a more quantitative comparison, we show in Fig. 5 the wind differences between BU and MerU, and their standard deviations and percentage differences. Here, the wind difference ($\Delta u_{im}$) at each height ($i$) and month ($m$) is calculated by subtracting the wind of other dataset ($u_{im}^{ot}$) from the BU ($u_{im}^{bu}$). At each height the percentage difference ($P_i$) is defined as the ratio of the standard derivations ($\sigma_i$) of $\Delta u_{im}$ to the peak BU.

At the equator (Fig. 2b and 2c), the agreements of BU and MerU are good at least below ~55 km and can be described as: (1)
at ~40-55 km, The SAO is dominant in both BU and MerU and have nearly identical amplitudes and phases, as well as zero wind lines; (2) at ~20-35 km, the QBO is dominant in both BU and MerU and have nearly identical zero wind lines. (3) at ~35-40 km, both SAO and QBO can be seen in BU and MerU; (4) both BU and MerU reproduce the disrupted QBO in 2016, i.e., a westward jet (highlighted by a red solid rectangle) formed within the eastward phase of QBO at 18-23 km. Meanwhile the eastward wind shift to a higher height (above 23 km) (Newman, et al., 2016; Osprey, et al., 2016; Diallo et al., 2018).
Subsequently, the westward wind (highlighted by a red dashed rectangle) occurred in the eastward phase of QBO in ~30-35 km during late 2016; (5) at ~40-50 km, the fast westward jet (denoted as white rectangles) seems an extension of the westward phase of QBO to a higher height just after the QBO changing its phase from eastward to westward. This feature can also be seen in Fig. 6 of Smith et al. (2017), which showed the BU derived from the SABER observations. The wind differences shown in Fig. 5a exhibit that BU is more westward (eastward) than MerU below ~30 km during the period of QBO westward (eastward)
phase. At z~30-55 km, BU is more westward than MerU with peak differences of ~20 ms$^{-1}$. Above ~55 km, the BU is more eastward than MerU with peak differences of ~60 ms$^{-1}$. A possible reason for the less eastward MerU is the strong damping of MERRA2 (Ern et al., 2021). The standard deviations of the wind differences (left column of Fig. 5a) are less than 7 ms$^{-1}$ below ~40 km and is about 10 ms$^{-1}$ above ~42 km. The large percentage differences (middle column of Fig. 5a) with magnitudes of ~30-40% occur at around 20 km and 43 km. In the other height ranges, the percentage differences are ~20%.
At 30°N/S (Fig. 3), the excellent agreements between BU and MerU can be summarized as the following three points: (1) at 30°S (Fig. 3a and 3b), the eastward jets are asymmetry around June with peaks at a lower (higher) height during early (later) summer of most years; (2) at 30°N (Fig. 3c and 3d), the eastward jets of both BU and MerU have two relative weaker peaks below ~60 km. Then with the increasing height, the two peaks merge into one strong peak above ~70 km during winter of most years; (3) both BU and MerU have nearly identical patterns of zero wind lines and westward wind jets. A more
quantitative comparisons shown in Figs. 5b and 5c exhibit that the wind differences exhibit asymmetric AO generally except for the short-term variations with periods of several months. The asymmetry means that the eastward phase of AO in the wind differences lasts a longer time than the westward phase. Comparisons between Figs. 5b and 3 show that the AO in wind difference is generally in phase with that in the zonal wind. This indicates that the BU is more eastward (westward) than MerU when the wind phase is eastward (westward). Compared to the wind differences before August 2004, the wind differences are
smaller above ~60 km. This might be a consequence of the improved quality of MERRA2 after assimilating the MLS data (Molod et al., 2015; Gelaro, et al., 2017). The standard deviations of the wind differences (left column of Figs 5b and 5c) vary from ~3 ms$^{-1}$ to ~8 ms$^{-1}$ with increasing heights. The percentage difference is ~10% in the entire height range, except for ~35% at 21 km and 30°S.

At 50°N/S (Fig. 4), the eastward jets of both BU and MerU are stronger at 50°S than those at 50°N below ~70 km. Moreover,
the westward jets of both BU and MerU reach their peaks at higher heights than those of the eastward jets. At 50°S, the BU
and MerU agree well with each other and can be described as the following four aspects: (1) the nearly identical patterns of
zero wind lines, (2) the fastest (slowest) eastward winds during 2006 (2010), (3) the nearly identical times (around July) and
heights (~50 km) of the eastward wind jets, (4) the nearly identical times (around January) and heights (~70 km) of the
westward wind jets. At 50°N, the good agreements between BU and MerU exhibit the four aspects mentioned above. Moreover,
the double-peak structure can be seen in both BU and MerU during winter of some years (e.g., 2003, 2004, 2006, 2008, 2009,
2012, 2013, 2015, 2019). These double-peak structures caused by the sudden stratospheric warmings (SSWs), which reduce
eastward wind during minor SSW or even reverse the eastward wind to westward during major SSW (Butler et al., 2015; 2017).
From Fig. 5d and 5e, we see that the wind differences exhibit asymmetric AO. Here the asymmetries mean that: (1) the time
interval of the eastward and westward phases are different in each hemisphere, and they are also different between 50°N and
50°S. At 50°S, the westward (eastward) phase lasts a longer time than the eastward (westward) phase below (above) ~60 km.
In contrast, the eastward phase lasts a longer time than the eastward phase at 50°N. The standard deviations of the wind
differences (left column of Figs. 5d and 5e) vary from ~3 ms$^{-1}$ to ~12 ms$^{-1}$ with the increasing height. The percentage difference
is in the range of 10-17% except for ~20% at 38 km and 50°N.

The above comparisons show that BU and MerU agree well with each other. At the equator and below 55 km, BU and MerU
have nearly identical zero wind lines and reproduce the QBO and SAO. Both BU and MerU reproduce the fast westward jet
during the beginning of the QBO westward phase and the disrupted QBO in early 2016. At middle and high latitudes, BU and
MerU have nearly identical zero wind lines in z=20-70 km and reproduce the year-to-year variations of the latitude and height
dependent eastward/westward jets. Both BU and MerU reproduce the double-peak structures induced by SSWs in the NH.

**3.2 Comparisons with the Winds from UARP and HWM14 in a Composite Year**

To compare the UraU with BU, the UarU is interpolated to geometric height with vertical interval of 1 km and latitude interval
of 2.5°. Moreover, the BU is rearranged in a composite year, which is calculated by averaging the BU in the same calendar
month of the years from 2002 to 2019. Fig. 6 shows the latitude-height sections of BU and UraU and their differences in each
month of the composite year. It shows that zero wind lines of BU and UraU are nearly identical below ~85 km from 50°S to
50°N. During winter (November, December, January, February) in the NH and summer (May, June, July, August) in the SH,
the eastward jet shifts from high to low latitudes with the increasing height. During summer in the NH and winter in the SH,
the westward jet shifts from low to high latitudes with the increasing height. During spring (March, April) and autumn
(September, October), the westward winds, which occur at low and middle latitudes and below ~45 km or above ~70 km, are
separated by eastward winds at ~45-70 km. These comparisons show the good agreement between BU and UraU below ~80
km.

Above 80 km and at middle to high latitudes, Both BU and UraU exhibit similar eastward jets during winter in the SH and
during summer in the NH. During summer in the SH and winter in the NH, both BU and UraU exhibit decreasing eastward

wind with the increasing height and even reverse to westward near 100 km. This is different from the MetU at MH (53.5°N) and BJ (40.3°N) (red lines in Fig. 8), in which the westward winds occur only around March and April. The comparisons among the BU, MetU and UraU will be shown in the next subsection.

At around the equator and above 80 km, both BU and UraU exhibit westward winds but have different height and latitudinal coverages. At ~90 km, the UraU is eastward throughout the composite year except during February-April. However, the BU is eastward only during June and December and westward during other months. By analyzing the zonal winds measured by the medium frequency (MF) radars at Christmas Island (2°N, 157°W), Tirunelveli (8.7°N, 77.8°E) and Pameungpeuk (7.4°S, 107.4E°) and meteor radar at Koto Tabang (0.20°S, 100.3°E) and Jakarta (6°S, 107°E), Rao et al. (2012) showed that the

composite zonal winds is westward except around June, July and December. This supports the BU derived here. Comparing between the BU calculated from Eq. (4) (Fig. 2a) and the reconstructed BU (Fig. 2b and 2c) above 80 km, we find that the BU calculated from Eq. (4) is eastward in general. While the reconstructed BU is largely westward and coincides with the radar observation results of Rao et al. (2012). Thus, the reliability of the BU at around the equator above 80 km is improved greatly after we combine the BU derived from the SABER observations and MetU measured by meteor radar at KT (0.20°S).

Although BU and UraU exhibit the general consistency in climatologic sense, there are still some differences between them. The wind differences between BU and UraU (the second and fourth rows of Fig. 6) reach eastward maxima of 30 ms$^{-1}$ at around the equator and at ~70-80 km during March-April and October-December. There are also westward differences with peaks larger than 15 ms$^{-1}$ at ~50 km and the equator during January-April. The westward peaks of -30 ms$^{-1}$ occur at around 30°N/S and at ~85-95 km during February-May and August-October. There are also westward differences, which extend

downward from ~ 100 km to ~50 km in the SH during November-January and in the NH during June-July. A short summary for the wind differences is that the eastward (westward) differences occur at around the equator (30°N/S).

Now, we compare the BU and HwmU. In a same manner as Fig. 6, we show in Fig. 7 the HwmU and BU in a composite year. Below ~85 km the eastward jets of BU and HwmU agree well during winter in the NH and during summer in the SH. Meanwhile, the westward jets of HwmU and BU agree well during summer in the NH and during winter in the SH. During

spring and autumn and at low and middle latitudes, the westward winds below ~45 km or above ~70 km surround the eastward winds at ~45-70 km. These comparisons show the good agreement between BU and HwmU below ~80 km.

At around the equator and at ~90 km, the eastward HwmU occurs during May-July and November-December, which lasts a shorter time interval than that of the UraU. At ~20-40°N, above the peaks of the eastward jets (at ~70 km), there are weak eastward jets in UraU (HwmU) during September-November (September-December) at 80-90 km with peak at ~85 km.

However, this peak is not as obvious as in BU or in the MetU at BJ (40.3°N) and SY (18.3°N) or in the LidU at CSU (40.6°N) shown in the next subsection.

The differences between BU and HwmU should be mentioned (the second and fourth rows of Fig. 7). Among the composite year, the eastward differences with peaks of ~45 ms$^{-1}$ occur at ~60-80 km during May-August in the SH. This is different from the wind differences between BU and UraU, in which the eastward differences reach their peaks at around the equator. The

westward differences with peaks of ~-30 ms$^{-1}$ occur at height of ~30-50 km and latitudes of 30-50°N (30-50°S) during winter

in the NH (SH). Moreover, the westward differences with peaks of $\sim$-30 ms$^{-1}$ occur at $\sim$55-75 km and latitudes of 30-50°S (30-50°N) during winter in the SH (NH). Above $\sim$80 km, the wind differences are westward in general throughout the composite year. This is in a situation like the wind difference between BU and UraU.

### 3.3 Comparisons with the Time Series of Winds Measured by Meter Radars

Fig. 8 shows the monthly mean zonal wind from meteor radars (MetU) and the BU at similar latitudes and five height levels. At MH (53.5°N) station, the variations of BU agree well with MetU with correlation coefficients (CCs) of 0.89-0.98. Moreover, BU and MetU exhibit similar AO and SAO above 90 km and similar AO below 86 km. The eastward peak of AO in both BU and MetU shift from below 86 km in winter to above 92 km in summer. From 2011 to 2019 and above 86 km, both BU and MetU show that the eastward wind is dominant in almost of all months except March and April, during which the MetU

reversed from eastward to westward. Here we note that the BU at 50°N, which represents the latitude range of 47.5°N-52.5°N, is near the location of MH (53.5°N) station. The slight difference of latitude might contribute some discrepancies between the BU and MetU at MH (53.5°N) station.

    At BJ (40.3°N) station, the variations of BU agree with MetU with CCs of 0.85-0.95. The AO and SAO of BU and MetU at BJ vary with height. Both BU and MetU show that the eastward peaks of AO shift from winter below 82 km to summer above

90 km. This is in a similar situation to that at MH station. At 86 km, the AO and SAO are almost equal portioned for the MetU with peaks in both summer and winter, while the AO is dominant with eastward peaks in winter. From 2009 to 2019 and above 86 km, both BU and MetU show that the eastward wind is dominant except in some occasional period.

    At SY (18.3°N) station, the CCs between BU and MetU are in the range of 0.78-0.9. However, the temporal variations of BU do not coincide well with that of MetU since the BU (MetU) is dominant by both AO and SAO (AO) above 86 km. Only at 82

345    km, the temporal variations of BU and MetU agree well with each other. Moreover, at 82 km, 86 km and 90 km, besides the AO and SAO, the QBO signal (with westward peaks in the beginning of 2011 and 2013) appears in both BU and MetU.

    At BK (1.2°S) station, the BU and MetU agree well with each other and have CCs of more than 0.88 below 86 km. The smaller CCs above 90 km are mainly caused by the inconsistency of weak temporal variations in BU and MetU. It should be noted that the magnitudes of BU agree well with those of MetU above 90 km. Moreover, both BU and MetU exhibit similar AO and

SAO from 82 to 98 km except at 94 km.

    At CP (22.7°S) station, the CCs between BU and MetU are in the range of 0.78-0.96. The temporal variations of BU and MetU agree well below 90 km. At 94 and 98 km, although the BU and MetU have CCs of 0.9 and 0.96, they are dominant by the SAO and AO, respectively. At SM (29.7°S) station, the CCs between BU and MetU are in the range of 0.82-0.95. The temporal variations of BU and MetU agree well with each other except at 86 km and 90 km. At 82, 94 and 98 km, both BU and MetU

are dominant by AO. While the AO in MetU has a lager amplitude than that of BU.

    A short summary about the comparisons among the time series of BU and MetU is below. At MH (53.5°N), BJ (40.3°N), BK (1.2°S) and SM (29.7°S) stations, the agreements between BU and MetU are good in general. The agreements are better at 82

km, 94 km and 98 km than those at 86 km and 90 km. At SY (18.3°N) station, the agreement between BU and MetU is good only at 82 km. At CP (22.7°S) station, the agreement between BU and MetU is good only below 90 km.

**3.4 Comparisons with the Winds Measured by Meteor Radars and Lidar in a Composite Year**

To compare BU, UraU, HwmU and MetU above 80 km in the terms of climatology, we show their monthly mean values and their differences in a composite year in Fig. 9. At 50°N and MH station (the first row of Fig. 9), the winds of the four datasets exhibit AO and agree well with each other below 85 km. The exception is that the BU and MetU are more westward than UraU and HwmU in summer and are less eastward than UraU and HwmU in winter. Above 86 km, the agreement is also good except that: (1) the westward wind in MetU during April does not occur in BU, UraU and HwmU; (2) the westward wind in UraU during December does not occur in BU, HwmU and MetU; (3) the SAO in BU and MetU cannot be seen in UraU and HwmU. The eastward differences of BU-MH with peaks larger than 15 ms$^{-1}$ occur above ~85 km and during April-June. In contrast, the eastward differences of larger than 15 ms$^{-1}$ last a longer time interval for UraU-MetU and HwmU-MetU. Moreover, the westward differences are also larger in UraU-MH and HwmU-MH than those in BU-MH.

At 40°N, and BJ and CSU station (the second row of Fig. 9), the BU exhibit AO and shift its eastward peak from below ~90 km in winter to above ~90 km in summer. In contrast, the AOs in MetU and LidU shift their eastward peaks below ~85 km in winter to above ~85 km in summer. The difference between MetU and LidU is that the westward LidU shift from ~80 km in June to ~100 km in March, while the westward MetU occurs only below ~85 km. The weak eastward BU and MetU extend upward from ~85 km in April and October ~100 km in January and December. This is different from those UraU and HwmU: (1) the eastward wind peaks of both UraU and HwmU are stronger than those of BU, MetU and LidU; (2) the weak eastward peaks at ~90 km in March and September do not appear in either BU or MetU or LidU. Thus, the BU at 40°N agrees with MetU and LidU better than that with UraU and HwmU. The westward differences of BU-BJ with peaks less than -15 ms$^{-1}$ occur below ~88 km and during May-July. In contrast, the differences of UraU-BJ and HwmU-BJ are eastward with peak values of lager than 15 ms$^{-1}$, which are larger than those of BU-BJ. The westward differences of BU-BJ and eastward differences of UraU-BJ and HwmU-BJ are responsible for the westward differences of BU-UraU and BU-HwmU at latitudes higher than 30°N (see Figs. 6 and 7). Comparisons among the differences of BU-BJ, UraU-BJ and HwmU-BJ, the magnitudes of the differences of BU-BJ are the smallest one, although they vary with month and height.

At 18.75°N and SY station (the third row of Fig. 9), the westward BU at ~80 km in March-May shifts upward to ~100 km in January-April. This agrees with the MetU except that the westward MetU lasts a shorter time interval. However, this is different to those of UraU and HwmU, which experience eastward wind at ~87-95 km. The westward BU at ~80 km in August shifts upward to ~100 km in October. In contrast, this westward BU does not occur in either UraU, HwmU or MetU. The eastward wind peaks during summer at ~80-100 km can be seen in the four data sets. The height range and time lasting of westward differences of BU-SY are lager and longer than those of UraU-SY and HwmU-SY. The peak value of the westward differences of BU-SY is more negative than -15 ms$^{-1}$. This contrasts with the eastward difference of 15 ms$^{-1}$ for UraU-SY and HwmU-SY.

At 1.25°S and BK station (the fourth row of Fig. 9), the agreement between BU and MetU is excellent on the aspects of both seasonal and height variations. This is mainly because the BU is reconstructed by the MetU at the KT station, which is at 0.20°S and is very near the BK station (1.18°S). The UraU and HwmU exhibit similar SAO with eastward peaks during summer and winter. These eastward peaks are much stronger than those of BU and MetU, especially below ~92 km. In contrast, above ~92 km, the UraU and HwmU are more westward than those of BU and BK. This is different from the results of Rao et

al. (2012), who showed that the UraU was less westward than the MetU at KT station (0.20°S). A possible reason is that the 3-month running mean is performed here to construct a smooth BU at the equator and reduces the peak magnitudes of zonal winds. The differences of BU-BK are smaller than those of UraU-BK and HwmU-BK. This might be the inclusion of the zonal wind measured at KT station (0.20°S) when we construct BU at the equator. In contrast, the differences of UraU-BK and HwmU-BK reach their peak values of larger than 15 ms$^{-1}$ in summer and winter.

At 22.5°S and CP station (the fifth row of Fig. 9), the agreement between BU and MetU is good below ~92 km. Both of them exhibit: (1) SAO with eastward peaks in summer and winter; (2) westward winds during September-October and shifting backward with height. However, this westward wind cannot be seen in either UraU or HwmU. Above ~92 km, the agreement among the four datasets is good during January-March and October-December. During May-September, the eastward wind in BU cannot be seen in the other three datasets. This might be a symmetry results as those at 18.75°N, where the eastward wind

in BU in November-December cannot be seen in the other three datasets. The differences of BU-CP are eastward with peak of more negative than -15 ms$^{-1}$ in summer and westward in winter with peak of larger than 15 ms$^{-1}$. In contrast, the differences of UraU-CP are eastward with peak of larger than 15 ms$^{-1}$ below ~92 km and westward above ~92 km. Among the four datasets, the differences of HwmU-CP are smallest in general. A possible reason is that the winds measured medium frequency radar at similar latitudes (Bribe Island at 28°S) have been included in the HWM14 model (Drob et al., 2008).

At 30.0°S and SM station (the sixth row of Fig. 9), the agreement between BU and MetU is good in general. Exception is that the BU is less eastward than the other three datasets during March-June at 82-95 km. Both BU and MetU exhibit much weak eastward winds than UraU and HwmU during September-November at ~82-95 km. During May-September and above ~92 km, the eastward wind in BU cannot be seen in the other three datasets. However, the eastward wind in BU is weaker and extends a shorter time interval as compared to that at 22.5°S and is closer to the MetU than those of UraU and HwmU. The

differences of BU-SM are mainly westward with peak of more negative than -15 ms$^{-1}$ during April. In contrast, the differences of UraU-SM are eastward with peaks of larger than 15 ms$^{-1}$ at ~82-92 km throughout the composite year. Although HWM14 model included the winds measured by the medium frequency radar at Adelaide (34.5°S) and Bribe Island at (28°S), there are still eastward differences with peaks of larger than 15 ms$^{-1}$ during June and September.

    A short summary of the comparisons among BU, UraU, HwmU, LidU and MetU in the composite year is below. The BU

agrees with MetU and LidU better than UraU and HwmU at 50°N, 40°N, 1.25°S and 30°S. At 18.75°N and 22.5°S, the agreement between BU and MetU is better than that among UraU, HwmU and MetU during spring and summer since the westward winds in BU and MetU cannot be seen in UraU or HwmU. However, the agreement between BU and MetU is worse than that among UraU, HwmU and MetU since the westward BU cannot be seen in the other three datasets in autumn. The

agreement between BU and MetU at 1.25°S indicates that the reconstructed BU is a feasible way get reliable zonal wind at the
425 equator above ~80 km. The agreement between BU and MetU at 50°N, 40°N and 30°S indicates that the reliability of balance wind theory can be extend up to the height of 100 km. The less agreement between BU and MetU at 18.75°N and 22.5°S indicates that the tidal aliasing to the mean wind might not be neglected at these latitudes.

## 4 Conclusions

Using the temperature and pressure observations by the SABER instrument over the past 18 years (2002-2019) and the balance
wind theory, we derive the monthly zonal mean zonal wind (BU) at 18-100 km and at 10°N-50°N and 10°S-50°S and at the equator. The BU at the equator and above 80 km is replaced by the zonal wind measured by the meteor radar at KT station (0.20°S). Therefore, the reconstruct BU overcomes the tide alias at 80-100 km. Then the cubic interpolation is applied to get BU at 7.5°S-7.5°N.

The BU is compared with the zonal winds from MERRA2 re-analysis data (MerU), UARP data (UraU), HWM14 empirical
model (HwmU), and meteor radar observations (MetU) at stations of MH (53.5°N), BJ (40.3°N), SY (18.3°N), BK (1.2°S), CP (22.7°S) and SM (29.7°S) and the lidar observations at CSU (40.6°N). The main conclusions can be summarized as the following:

The comparisons between BU and MerU show good consistency. At middle and high latitudes, BU and MerU have nearly identical zero wind lines in 18-70 km and year-to-year variations of the eastward/westward wind jets, the double-peak
structures caused by SSW. At the equator and below 55 km, BU and MerU have nearly identical zero wind lines and reproduce the QBO and SAO, especially the disrupted QBO in early 2016.

The comparisons among BU, UraU and HwmU show good agreement in general below 80 km. They have nearly identical zero wind lines and reproduce the eastward (westward) jet in the winter (summer) hemisphere. Above 80 km, the good agreement among BU, UraU and HwmU is at the high latitudes.

The comparisons among BU, UraU, HwmU, MetU and lidU show that the BU agrees with MetU and LidU better than UraU and HwmU at 50°N, 40°N, 1.25°S and 30°S on the aspects of both their time series and composite year. At 18.75°N and 22.5°S, the time series of BU agrees with those of MetU only at 82 km and below 90 km, respectively. In a composite year, the BU agrees with MetU better than UraU and HwmU in spring and summer.

Based on the comparisons, we conclude that the BU derived here is reliable at 18-100 km and from 50°S to 50°N in general.
The BU derived here covers a time span of 18 years and can be used to study the seasonal and interannual variations (e.g., SAO, AO, QBO, ENSO etc.) as well as their global interactions from the stratosphere to the lower thermosphere. It can also serve as the background for wave (such as, gravity waves, tides and planetary waves) propagations and global interactions from the stratosphere to the lower thermosphere. The BU data have been archived as netCDF files and can be available at National Space Science Data Center through https://dx.doi.org/10.12176/01.99.00574.

***Data availability.*** The SABER data are downloaded from ftp://saber.gats-inc.com/Version2_0/Level2A/ (last access: March 2020). The UARP wind data were obtained from ftp://sparc-ftp1.ceda.ac.uk/sparc/ref_clim/randel/temp_wind (last access: March 2020). The MERRA2 data were obtained from were obtained from http://disc.sci.gsfc.nasa.gov/mdisc (last access: March 2020). The meteor radar data of MH, BJ and SY were supported by the Chinese Meridian Project (Wang, 20) and provided by Beijing National Observatory of Space Environment, Institute of Geology and Geophysics Chinese Academy of Sciences through the Geophysics center, National Earth System Science Data Center (http://wdc.geophys.ac.cn, last access: March 2020). The meteor radar data of KT and BK were provided by Research Institute for Sustainable Humanosphere, Kyoto University (http://database.rish.kyoto-u.ac.jp/arch/iugonet, last access: March 2020). The database has been arranged by the Inter-university Upper atmosphere Global Observation NETwork (IUGONET) project (Hayashi et al., 2013). The meteor radar data at CP and SM are available upon request to Paulo Prado Batista. The CSU lidar data are available upon request to Tao Yuan. The BU data developed in this work can be available at https://dx.doi.org/10.12176/01.99.00574 (Liu et al., 2021).

***Author contribution.*** XL and JX designed the study and wrote the manuscript. JY contributed to the discussion of the results and the preparation of the manuscript. All authors discussed the results and commented on the manuscript at all stages.

***Competing interests.*** The authors declare that they have no conflict of interest.

***Acknowledgments.*** This work was supported by the National Natural Science Foundation of China (41831073, 41874182, 42174196), the Natural Science Foundation of Henan (212300410011), the Open Research Project of Large Research Infrastructures of CAS "Study on the interaction between low/mid-latitude atmosphere and ionosphere based on the Chinese Meridian Project". This work was also supported in part by the Specialized Research Fund and the Open Research Program of the State Key Laboratory of Space Weather.

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

**Table and Figures Captions**

**Table 1. Locations of radar and lidar and data periods**

| Instrument | Station | Location | Period | References |
|---|---|---|---|---|
| Meteor radar | Mohe (MH) | 53.5°N, 122.3°E | 2011-2019 | Li et al. (2012) ; Xiong et al. (2013); Yu et al. (2013, 2015) |
| Meteor radar | Beijing (BJ) | 40.3°N, 116.2°E | 2009-2019 | |
| Meteor radar | Sanya (SY) | 18.3°N, 109.6°E | 2011-2016 | |
| Meteor radar | Koto Tabang (KT) | 0.2°S, 100.3°E | 2002-2017 | Batubara et al. (2011); Rao et al. (2011, 2012); Hayashi et al. (2013); Abe et al. (2014); Matsumoto et al. (2016) |
| Meteor radar | Biak (BK) | 1.2°S, 136.1°E | 2011-2015 | |
| Meteor radar | Cachoeira Paulista (CP) | 22.7°S, 45.0°W | 2004-2008 | Batista et al. (2004); Andrioli et al. (2009, 2013, 2015) |
| Meteor radar | Santa Maria (SM) | 29.7°S, 53.8°W | 2005-2008 | |
| Na Lidar | Colorado State University (CSU) | 40.6°N, 105.1°W | 2002-2008 | She et al. (2004); Yuan et al. (2008) |

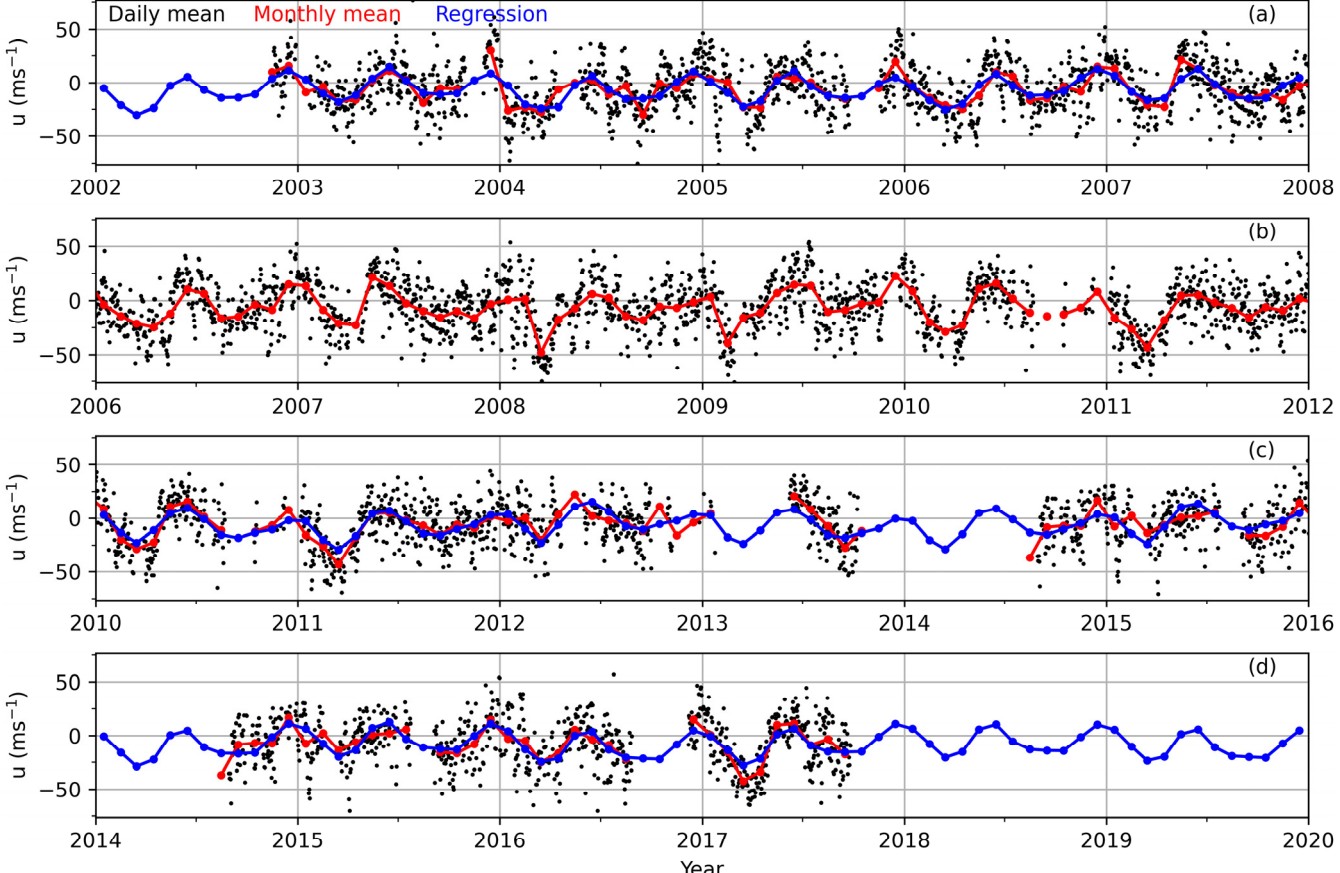

**Figure 1.** The daily mean (black dots) and monthly mean (red line with dots) zonal winds (positive for eastward) at 86 km measured by the JPNKT meteor radar and their regression results (blue line with dots) from 2002 to 2019. The x-ticks mark the beginning of each year.

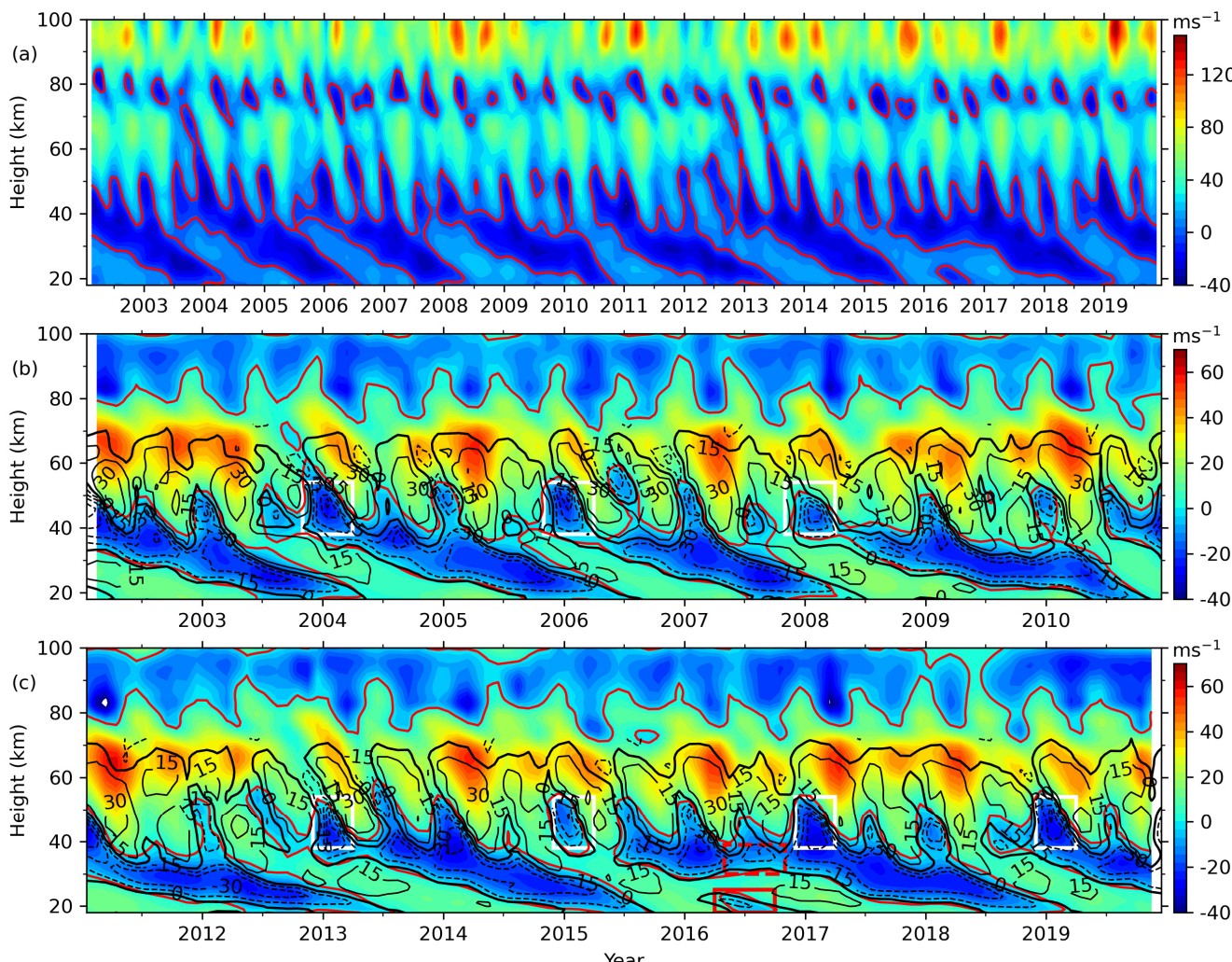

**Figure 2.** The BU (color filled contour, positive for eastward) calculated by Equation (4) at the equator at 18-100 km (a: 2002-2019) and the reconstructed BU (b: 2002-2010, c: 2011-2019). The overplotting contour lines are MerU (interval of 15 ms$^{-1}$, the eastward and westward winds are represented as solid and dash lines, respectively). The white rectangles highlight the fast westward jet during the beginning of the QBO westward phase. The red and thick black contour lines are the zero winds of BU of MerU, respectively. The red solid and dash rectangles show the disrupted QBO in 2016. The x-ticks mark the beginning of each year.

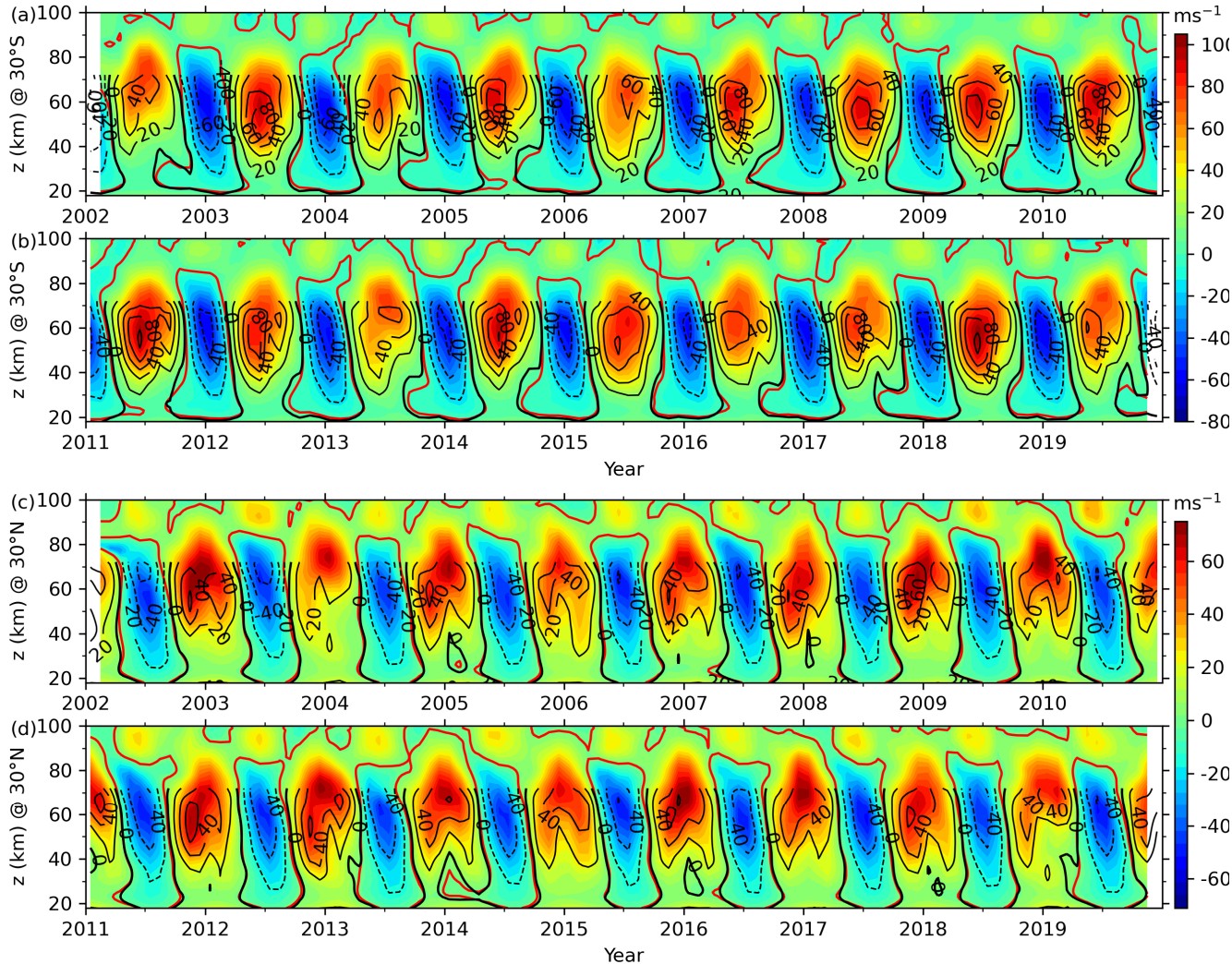

**Figure 3.** Time-height sections of the BU (color filled contour, positive for eastward) and MerU (lines with contour interval of 20 ms$^{-1}$, the eastward and westward winds are represented as solid and dash lines, respectively) at 30°S (a, b) and 30°N (c, d). The red and thick black contour lines are the zero winds of BU of MerU, respectively. Different color scales are used at 30°S and 30°N. The x-ticks mark the beginning of each year.

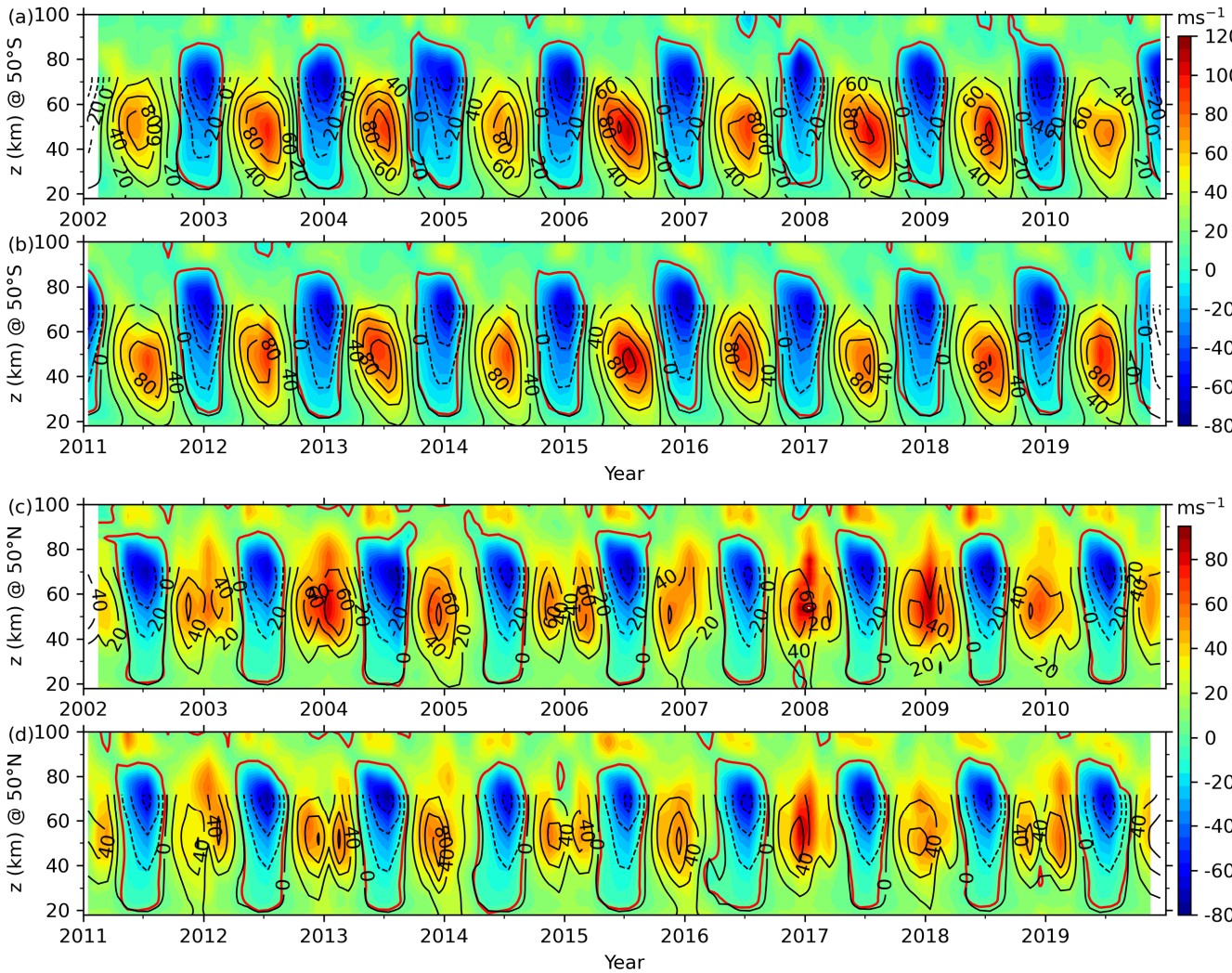

**Figure 4.** Same caption as Figure 3 but at 50°S (upper two panels) and 50°N (lower two panels).

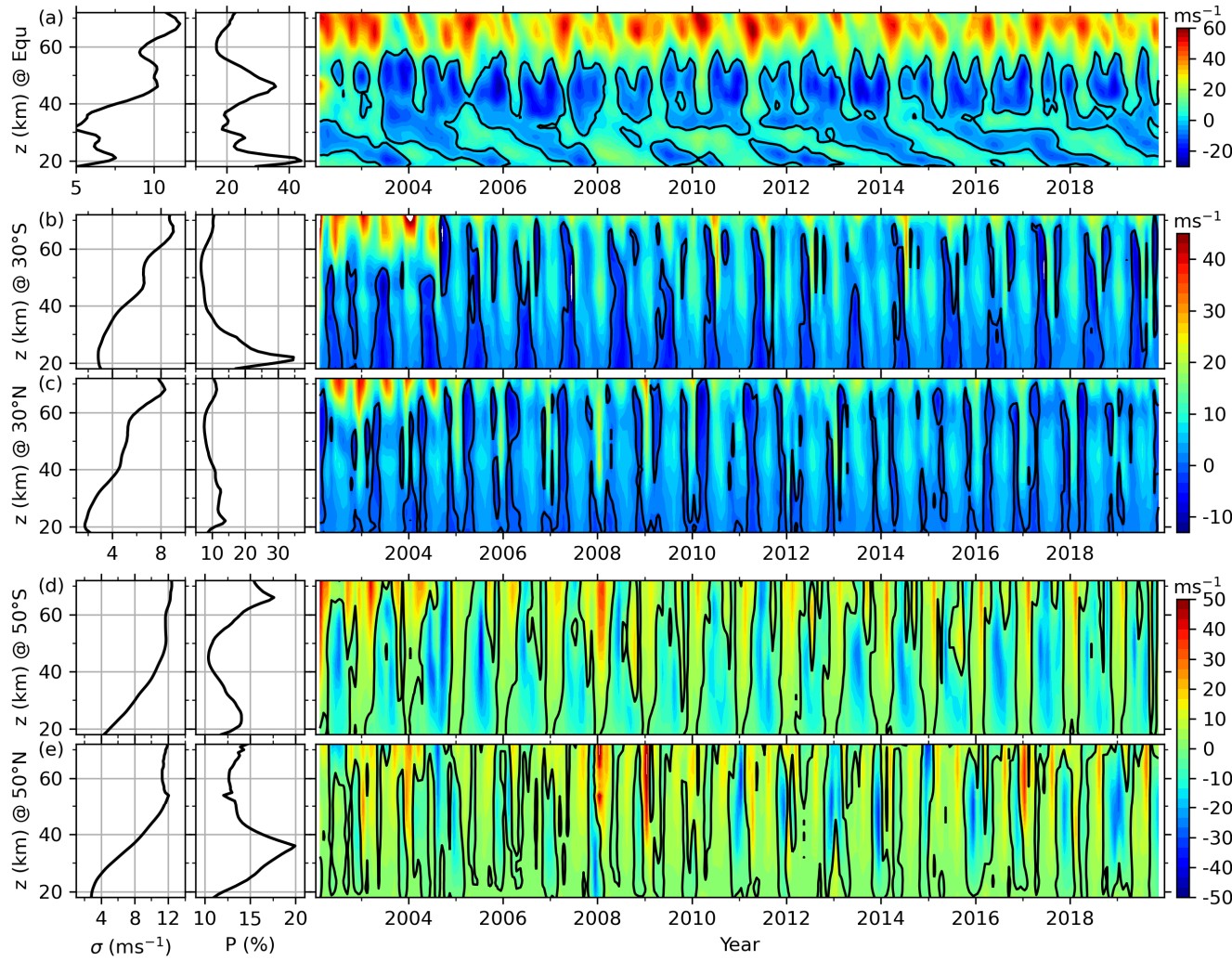

**Figure 5.** The wind differences between BU and MerU (right column) and their standard deviations ($\sigma$, left column) and percentage differences ($P$, middle column) at the equator (a), 30°N/S (b, c), and 50°N/S (d, e). The black contour lines are the zero wind difference.

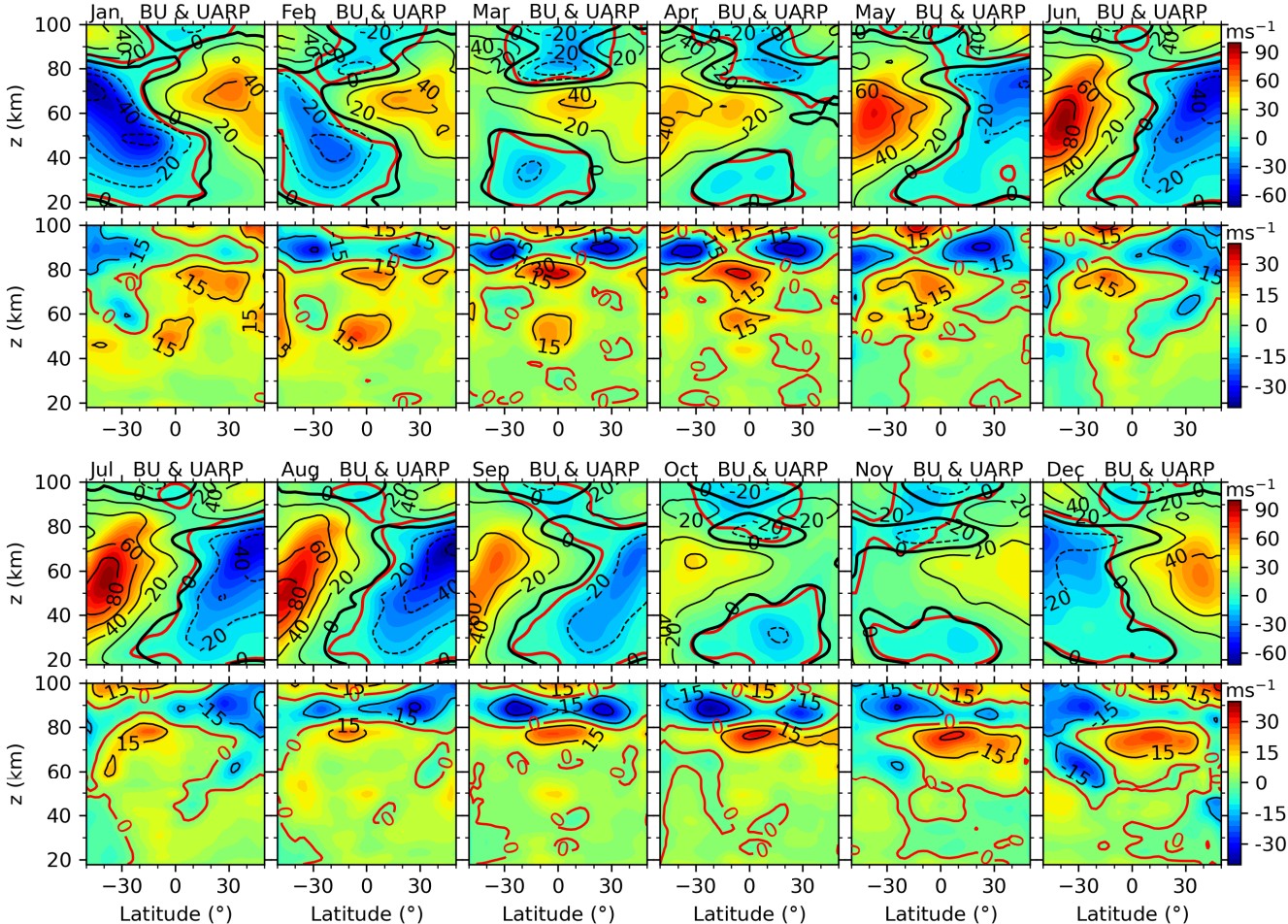

**Figure 6.** Latitude-height sections of BU (color filled contour, positive for eastward) and UraU (contour lines with interval of 10 ms$^{-1}$, the eastward and westward winds are represented as solid and dash lines, respectively) in each month (denoted one the left-upper corner of each panel) of a composite year. The thick black and red contour lines are the zero wind of UraU and BU, respectively. Same color scale is used for all months.

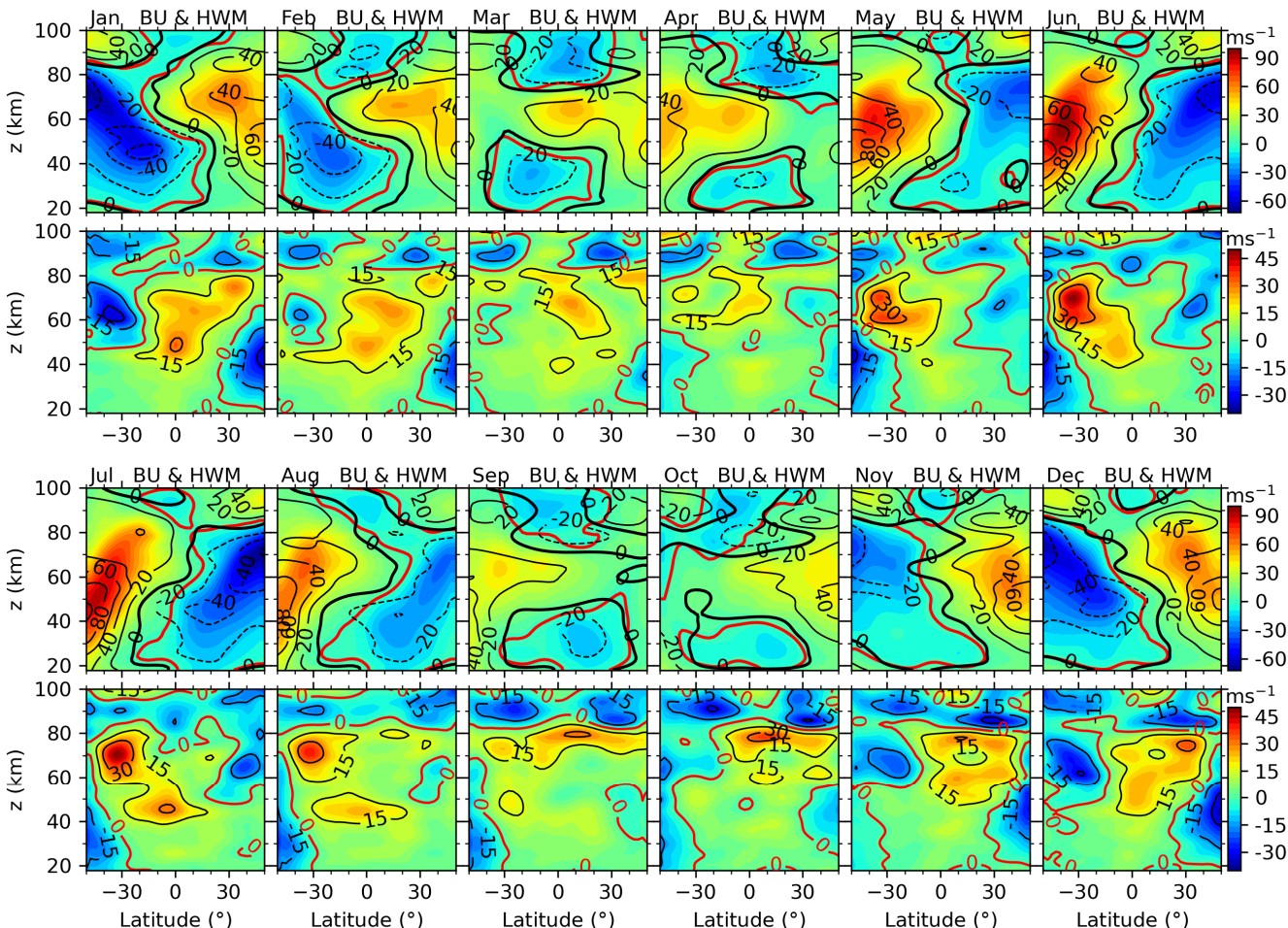

**Figure 7.** Same caption as Figure 5 but for the BU and HWM14. The thick black and red contour lines are the zero wind of HwmU and BU, respectively.

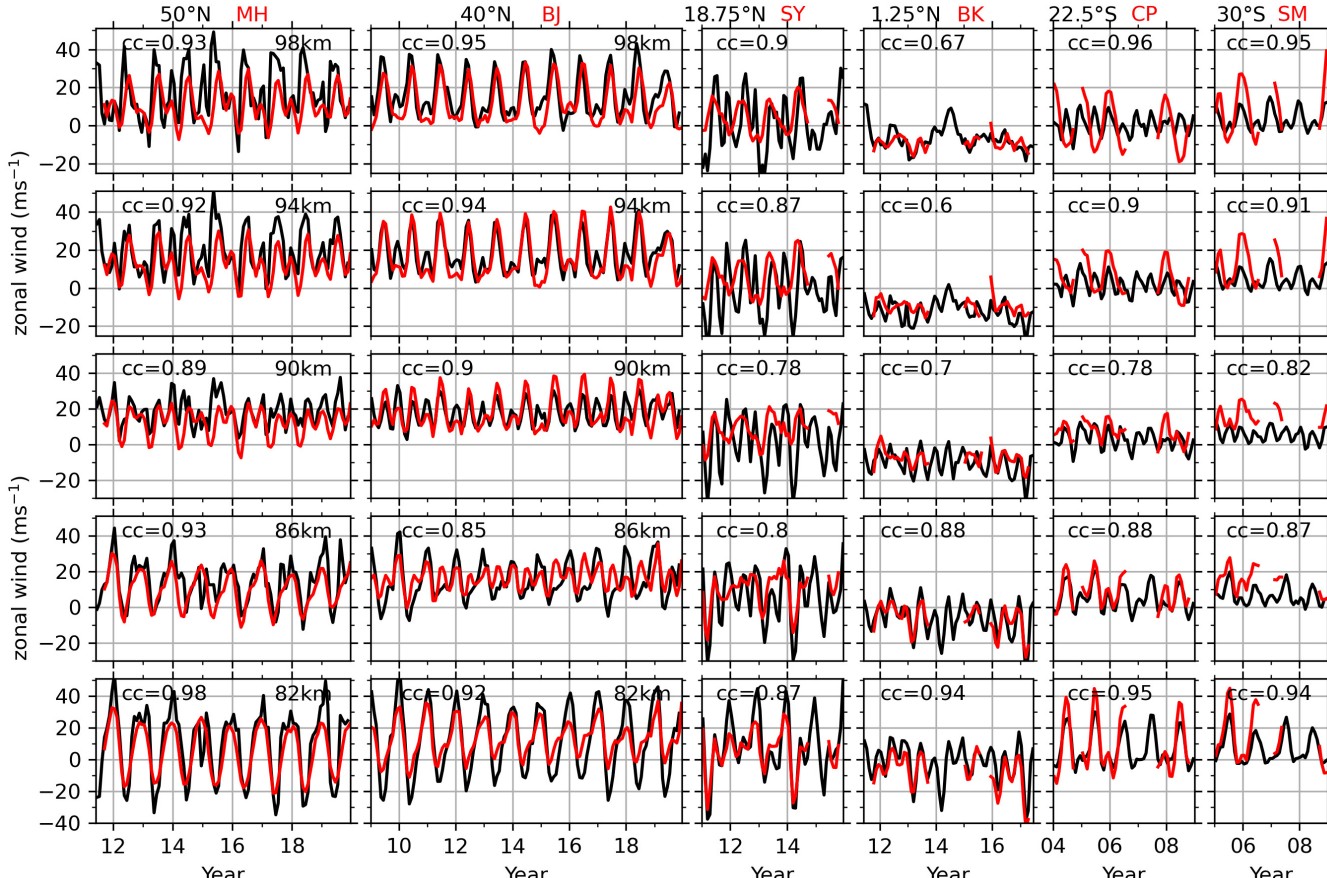

**Figure 8**. Monthly mean zonal wind from meteor radars (red lines, positive for eastward) at stations (from left to right) of MH (53.5°N), BJ (40.3°N), SY (18.3°N), BK (1.2°S), CP (22.7°S) and SM (29.7°S) and the BU (black) at the similar latitude (labeled on the top of each column) at five heights. The correlation coefficient (cc) between BU and MetU is labeled on each panel. Same y-axis is used in each row. The x-ticks mark the beginning of each year.

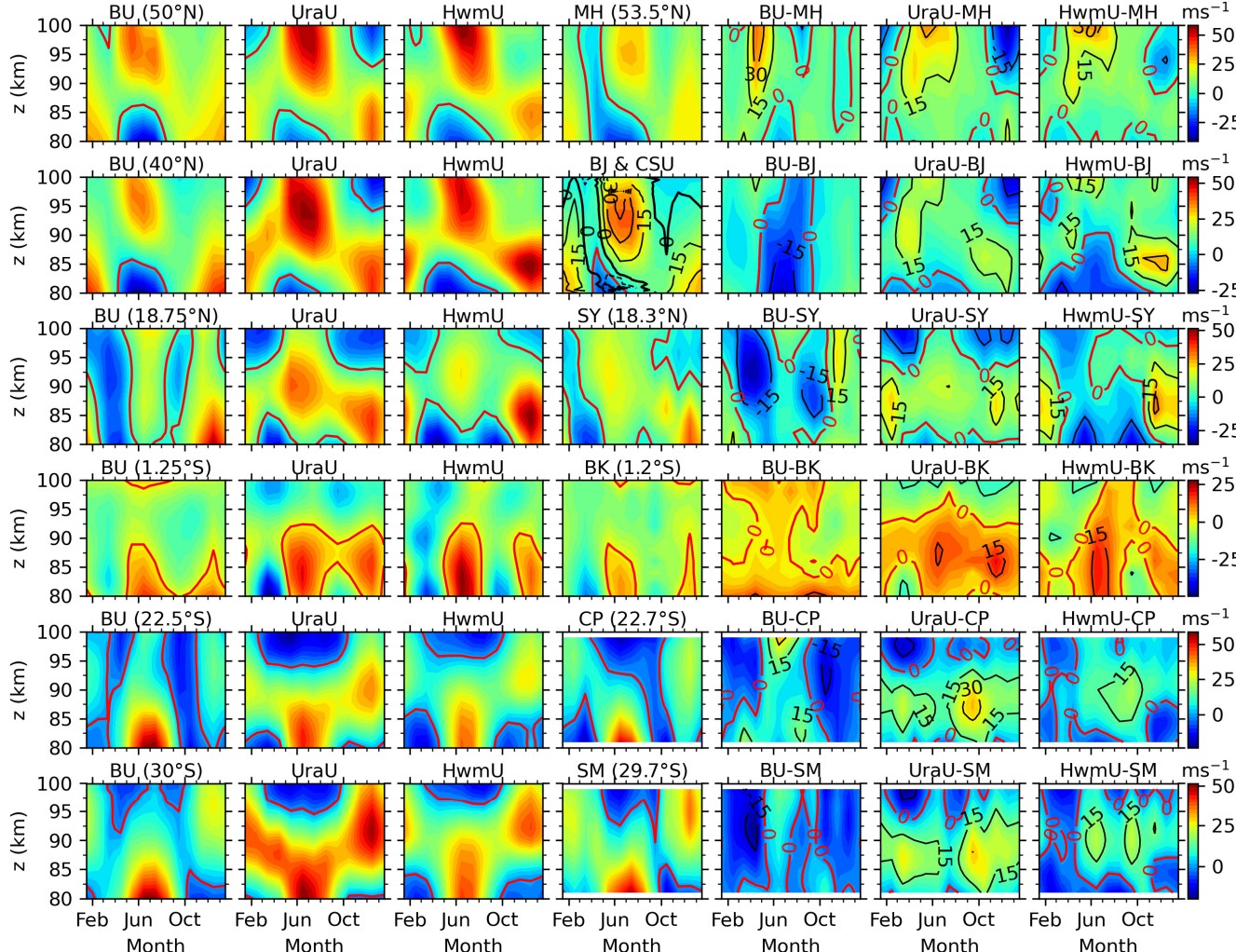

**Figure 9.** Monthly mean zonal winds of BU, UraU, HwmU, and MetU (from the first to fourth column) and their differences (from the fifth to seventh column) at stations of (from up to below) MH (53.5°N), BJ (40.3°N), SY (18.3°N), BK (1.2°S), CP (22.7°S) and SM (29.7°S) in a composite year. The red contour lines show the zero wind in each panel. The black contour lines (interval of 10 ms-1) show the zonal wind measured by the CSU lidar (LidU). The wind difference is represented by color filled contour and highlighted by contour lines.