# Peer review of "Global Balanced Wind Derived from SABER Temperature and Pressure Observations and its Validations"

_Earth System Science Data, 2021_

## Referee Comment (RC1)

**General Comments:** In my opinion, this paper is an instrumental compilation for researchers working on this topic, but it will also be of value to workers not very familiar with the details of the topic. The way the methods, results, and interpretations are presented is concise but provides sufficient details on the individual aspects. It is straightforward to follow the authors' reasoning and the explanations for their approach, and the implications for future research. Compilations like this one will always be crucial and help make data more approachable to researchers. Following the guidelines provided for reviewers, the data is of high quality. The paper is appropriate to explain the database provided. The length and structure of the article are appropriate, the language is consistent and precise, and the figures and tables are correct and of high quality. Overall, I would rate this manuscript "excellent" and think it deserves publication.

**Specific Comments:** The only specific comment I have is related to chapter 2.3. In lines 154-169, the wording could be improved. The continuous wind dataset is constructed based on the predictor variables, which are calculated by the multiple linear regression by considering the predictors, including constant of 1, the wind data in segment b, oscillations with periods of 12, 6, 36, 24, 4, and 3 months. It would be better to discuss the ability of the MLR method to capture the variation of wind, which could improve the rationality of data processing methods. For instance, one may wonder how much of the variation in the raw data in each segment could be explained by the

reconstructed data based on the MLR predictors.

**Technical Corrections:**

A short list of typos/inconsistencies:

L. 167: The sentence should be rewritten.

L. 207-208: "can be seen both in both BU and MerU" -->   "can be seen in both BU and MerU".

---

## Referee Comment (RC2)

**Review of "Global Balanced Wind Derived from SABER Temperature and Pressure Observations and its Validations"**

BY X. LIU ET AL.

**General remarks**

This paper describes and validates a new dataset of the monthly mean zonal wind in the height range of 18-100 km at latitudes of 50°S-50°N from 2002 to 2019, which is based on data measured by the SABER instrument. This constitutes a valuable contribution to ESSD. However, the paper is not suitable for publication in ESSD in its present form.

I think the paper could be better in three main points

- The paper should convince the reader why the BU data set should be used. To play devil's advocate, one might be tempted to conclude 'let us just use MERRA'. I know there are arguments (like the altitude range of the various data sets) but these arguments could be made much clearer.

- There is an extensive comparison in the paper between BU and other data sets, which is very good and helpful. However, such comparisons are much more helpful if conducted in a more quantitative way, rather than saying 'good agreement' or 'almost the same zero wind line'. I suggest analysing (and perhaps showing) actual difference plots and percentage differences.

- The theoretical basis for developing the BU data set is described in sect. 2.2. This description should be clear and straightforward to follow, which is not the case in its present form (see below).

I suggest to return the manuscript to the authors for major revisions.

**Comments in detail**

**Section 2.2**

The method of deriving the BU data set is discussed here; this is an important part of the paper. It needs to be clear and should be understandable (in principle) without going back to the cited literature. First Eq. (3) should be valid at the equator (as it can be simplified to Eq. (4) at the equator). Is this correct? But in line 131 you say 10°-50°N/S fot the BU data set – this seems to be a contradiction. Further, in l. 136, you say that Eq. (3) is valid from 8°S to 8°N as well as 70°-90°N and 70°-90°S. This is how I read your text. This is inconsistent with the given range of the BU data. I might not be correct here, but this discussion is not as clear as it should be.

**Comparison with reanalyses**

I would be helpful to know if any of the data used for BU are assimilated in MERRA2. Also be clearer about for which latitude range you compare MERRA2 and BU. Why do you not compare with ERA5? (And add the reference to Hersbach et al. (2020) for ERA5).

**References**

The citations are okay, but there could be a bit more recent references to scientific issues to which the data set could be applied. For example, Diallo et al. (2018) find that the QBO disruption in 2015-2016 reversed the lower stratosphere moistening triggered by the alignment of the warm ENSO event with westerly QBO in early boreal winter. Would the BU data set also be useful for ENSO?

Moreover, Ern et al. (2021) find that reanalyses reproduce some basic features of the SAO gravity wave driving and that higher-top models (ERA-5 and MERRA-2) show stronger gravity wave driving of the SAO eastward phase in the stratopause region and in the lower mesosphere. But reanalyses are limited by model-inherent damping in the upper model levels. Would such findings be relevant for the data set discussed here? You do not need to consider the specific papers/findings mentioned here, but they might be a starting point.

**Presentation**

Overall the paper is well written, but I suggest a revision to correct several small grammatical errors. In particular, get the difference between "well" (adverb) and "good" (adjective) correct.

**Minor Points**

- p 1, l. 20: 'tide alias' will not be clear to everyone, rephrase

- p. 1 l 23: make sure to clarify that (e.g.) the MERRA comparison is not only 53.3° to 29.7°. Also the data set is only 50°S-50°N, so how can you compare at 53.3°?

- p.1 l 25: I would not call the QBO in 2016 "anormal", I am not even sure if this is proper English. See for example Diallo et al. (2018).

- p.2, l. 57: Be specific about ECMWF: do you mean ERA5 or ERA-Interim or both? I guess you mean ERA5. Add the reference to Hersbach et al. (2020) for ERA5.

- p 3, l. 80: justify the choice of these latitudes.

- p. 4, l 113: should be 'Remsberg'

- p. 4, l 117: The original profiles are from SABER – correct? Be specific here.

- p. 5, l 147: I suggest to refrain from such abbreviations in titles

- 11, l. 333: "tide alias" is not clear without further explanation.

**References**

Diallo, M., Riese, M., Birner, T., Konopka, P., Müller, R., Hegglin, M. I., Santee, M. L., Baldwin, M., Legras, B., and Ploeger, F.: Response of stratospheric water vapor and ozone to the unusual timing of El Niño and the QBO disruption in 2015–2016, Atmos. Chem. Phys., 18, 13 055–13 073, https://doi.org/10.5194/acp-18-13055-2018, URL https://www.atmos-chem-phys.net/18/13055/2018/, 2018.

Ern, M., Diallo, M., Preusse, P., Mlynczak, M. G., Schwartz, M. J., Wu, Q., and Riese, M.: The semiannual oscillation (SAO) in the tropical middle atmosphere and its gravity wave driving in reanalyses and satellite observations, Atmos. Chem. Phys. Discuss., https://doi.org/https://doi.org/10.5194/acp-2021-190, in review, 2021.

Hersbach, H., Bell, B., Berrisford, P., Hirahara, S., Horànyi, A., Muñoz Sabater, J., Nicolas, J., Peubey, C., Radu, R., Schepers, D., Simmons, A., Soci, C., Abdalla, S., Abellan, X., Balsamo, G., Bechtold, P., Biavati, G., Bidlot, J., Bonavita, M., De Chiara, G., Dahlgren, P., Dee, D., Diamantakis, M., Dragani, R., Flemming, J., Forbes, R., Fuentes, M., Geer, A., Haimberger, L., Healy, S., Hogan, R. J., Hólm, E., Janiskovà, M., Keeley, S., Laloyaux, P., Lopez, P., Lupu, C., Radnoti, G., de Rosnay, P., Rozum, I., Vamborg, F., Villaume, S., and Thépaut, J.-N.: The ERA5 global reanalysis, Q. J. R. Meteorol. Soc., 146, 1999–2049, https://doi.org/10.1002/qj.3803, 2020.

---

## Author Comment (AC1)

**Responses to RC1:**

**General Comments**

In my opinion, this paper is an instrumental compilation for researchers working on this topic, but it will also be of value to workers not very familiar with the details of the topic. The way the methods, results, and interpretations are presented is concise but provides sufficient details on the individual aspects. It is straightforward to follow the authors' reasoning and the explanations for their approach, and the implications for future research. Compilations like this one will always be crucial and help make data more approachable to researchers. Following the guidelines provided for reviewers, the data is of high quality. The paper is appropriate to explain the database provided. The length and structure of the article are appropriate, the language is consistent and precise, and the figures and tables are correct and of high quality. Overall, I would rate this manuscript "excellent" and think it deserves publication.

**Response:** Thanks for your careful reading and comments. Your comments are valuable in improving the quality of our manuscript.

**Specific Comments**

The only specific comment I have is related to chapter 2.3. In lines 154-169, the wording could be improved. The continuous wind dataset is constructed based on the predictor variables, which are calculated by the multiple linear regression by considering the predictors, including constant of 1, the wind data in segment b, oscillations with periods of 12, 6, 36, 24, 4, and 3 months. It would be better to discuss the ability of the MLR method to capture the variation of wind, which could improve the rationality of data processing methods. For instance, one may wonder how much of the variation in the raw data in each segment could be explained by the reconstructed data based on the MLR predictors.

**Response:** Thanks for your suggestion. To clarify all the predictor variables, we have added "The predictor variables can be summarized as a constant of 1, the wind data in segment b, oscillations with periods of 36, 24, 12, 6, 4, and 3 months." in the end of step (2).

Following your suggestion, to quantify the rationality of the MLR method, we used $R^2$ score, which is the ratio of the variations in the observation data explained by the model and defined as,

$$R^2 = 1 - \sum_{i=1}^{n}(y_i - f_i)^2 / \sum_{i=1}^{n}(y_i - \bar{y})^2, \ \bar{y} = \frac{1}{n}\sum_{i=1}^{n} y_i. \tag{R1}$$

Here, $y_i$ and $f_i$ are the observation data and model results, respectively. The best $R^2$ score is 1 when the predicted values are the same as the observation data. For segments a, c, and d, their $R^2$ scores are 0.63, 0.59, and 0.65, respectively. And their available observation months are 60, 57, and

34, respectively. It should be noted that the $R^2$ scores increase with the increasing number of predictor variables. However, the increasing number of predictor variables reduces the robustness of the model when the available observation months are short (e.g., segment c). Thus, the predictor variables chosen here are an optimal compromise between the $R^2$ score and the robustness of MLR model.

This has been added to in the text to quantify the rationality of the MLR model.

**Technical Corrections**

A short list of typos/inconsistencies:

L. 167: The sentence should be rewritten.

**Response:** This sentence has been rewritten as "It is reasonable to expect that the MLR predictions in the time intervals of missing observations are reliable (e.g., 2013 and 2014, before November 2002 and after September 2017) and can be used to construct BU. "

L. 207-208: "can be seen both in both BU and MerU" --> "can be seen in both BU and MerU".

**Response:** We have revised it as "can be seen in both BU and MerU".

---

## Author Comment (AC2)

**Responses to RC2:**

**General Remarks**

This paper describes and validates a new dataset of the monthly mean zonal wind in the height range of 18-100 km at latitudes of 50°S-50°N from 2002 to 2019, which is based on data measured by the SABER instrument. This constitutes a valuable contribution to ESSD. However, the paper is not suitable for publication in ESSD in its present form.

I think the paper could be better in three main points

1. The paper should convince the reader why the BU data set should be used. To play devil's advocate, one might be tempted to conclude "let us just use MERRA" I know there are arguments (like the altitude range of the various data sets) but these arguments could be made much clearer.

2. There is an extensive comparison in the paper between BU and other data sets, which is very good and helpful. However, such comparisons are much more helpful if conducted in a more quantitative way, rather than saying "good agreement" or "almost the same zero wind line". I suggest analysing (and perhaps showing) actual difference plots and percentage differences.

3. The theoretical basis for developing the BU data set is described in sect. 2.2. This description should be clear and straightforward to follow, which is not the case in its present form (see below).

I suggest to return the manuscript to the authors for major revisions.

**Response:** Thanks a lot for your efforts in evaluating our manuscript. Your comments and suggestions are valuable for us to improve the quality of our manuscript. The point-to-point responses are below.

**Point 1.** We have added some advantages of BU dataset as comparing to reanalysis data and observations in the abstract, introduction, and conclusions.

In the end of the abstract, we have added "The advantages of the global BU dataset are the large vertical extent (from the stratosphere to the lower thermosphere) and long-term duration (2002-2019). The BU data is useful to study the temporal variations with periods ranging from seasons to decades at 50°S-50°N. It can be used as the background wind for atmospheric wave propagation."

In the last third paragraph of the introduction, we have added "In the current state, the direct global measurement of zonal wind in the upper stratosphere and mesosphere is difficult, and the model-inherent damping in the upper model levels of MERRA2 and ERA5 is still a challenge to get realistic wind in the mesosphere and lower thermosphere (MLT) region (Ern et al., 2021). A candidate

is combining the observations of temperature and pressure with balance wind theory to get zonal wind in the MLT region."

At the end of the introduction, we have added a paragraph to describe the possible applications of the BU dataset. "The advantages of the global BU dataset are their large vertical extent and long-term temporal coverage. The vertical extent is from the stratosphere to the lower thermosphere. The temporal coverage is from 2002 to 2019. Thus, the BU dataset can be used to study the global variations of zonal wind in time scales ranging from seasons to decades and from the stratosphere to the lower thermosphere. These variations include SAO, AO, QBO and ENSO (El Niño–Southern Oscillation, periods of 2-8 years, Baldwin and O'Sullivan, 1995). Although QBO and ENSO are originated from the lower atmosphere or sea surface, their influences are global and can extend to the stratosphere or even higher heights and latitudes (Baldwin and O'Sullivan, 1995; Baldwin et al., 2001). Moreover, the interactions among SAO, AO, QBO and ENSO are also important in modulating atmospheric waves, and composition from the stratosphere to the lower thermosphere (e.g., Xu et al., 2009; Liu et al., 2017; Diallo et al., 2018; Ern et al., 2011, 2014, 2021; Kawatani et al., 2020)."

At the end of the conclusion, we have revised as "The BU derived here covers a time span of 18 years and can be used to study the seasonal and interannual variations (e.g., SAO, AO, QBO, ENSO etc.) as well as their global interactions from the stratosphere to the lower thermosphere. It can also serve as the background for wave (such as, gravity waves, tides and planetary waves) propagations and global interactions from the stratosphere to the lower thermosphere."

**Point 2.** Following your suggestion, the comparisons are performed by analyzing difference plots and the percentage difference. The wind difference ($\Delta u_{im}$) at each height ($i$) and month ($m$) is calculated by subtracting the wind of other dataset ($u_{im}^{ot}$) from the BU ($u_{im}^{bu}$). At each height the percentage difference ($P_i$) is defined as the ratio of the standard derivations ($\sigma_i$) of $\Delta u_{im}$ to the peak BU. We have added a new figure (Fig. 5 of this version), which summarizes the wind differences, standard deviations ($\sigma$) and percentage differences ($P$) between BU and MerU shown in Figs. 2-4. Moreover, we have revised Figs. 5, 6, 8 of the last version (Fig. 6, 7, 9 of this version) by adding the wind differences. Such that the comparisons are in a more quantitative way. The following analysis is based on the figure numbers of this version and has been added in the text.

**The comparisons between BU and MerU**

The right column of Fig. 5a shows that the BU is more westward (eastward) than MerU below ~30 km during the period of QBO westward (eastward) phase. At z~30-55 km, BU is more westward than MerU with peak differences of ~20 ms$^{-1}$. Above ~55 km, the BU is more eastward than MerU with peak differences of ~60 ms$^{-1}$. A possible reason for the less eastward MerU is the strong damping

of MERRA2 (Ern et al., 2021). The standard deviations of the wind differences (left column of Fig. 5a) are less than 7 ms$^{-1}$ below ~40 km and is about 10 ms$^{-1}$ above 42 km. The large percentage differences (middle column of Fig. 5a) with magnitudes of ~30-40% occur at around 20 km and 43 km. In the other height ranges, the percentage differences are ~20%.

[Figure]

**Figure 5.** The wind differences between BU and MerU (right column) and their standard deviations ($\sigma$, left column) and percentage differences ($P$, middle column) at the equator (a), 30°N/S (b, c), and 50°N/S (d, e). The black contour lines are the zero wind difference.

At 30°N/S, the wind differences exhibit asymmetric AO generally except for the short-term variations with periods of several months. The asymmetry means that the eastward phase of AO in the wind differences lasts a longer time than the westward phase. Comparisons between Figs. 5b and 3 show that the AO in wind difference is generally in phase with that in the zonal wind. This indicates that the BU is more eastward (westward) than MerU when the wind phase is eastward (westward). Compared to the wind differences before August 2004, the wind differences are smaller above ~60 km. This might be a consequence of the improved quality of MERRA2 after assimilating the MLS data (Molod et al., 2015; Gelaro, et al., 2017). The standard deviations of the wind differences (left

column of Figs 5b and 5c) vary from ~3 ms$^{-1}$ to ~8 ms$^{-1}$ with increasing heights. The percentage difference is ~10% in the entire height range, except for ~35% at 21 km and 30°S.

At 50°N/S, the wind differences exhibit asymmetric AO. Here the asymmetries mean that: (1) the time interval of the eastward and westward phases are different in each hemisphere, and they are also different between 50°N and 50°S. At 50°S, the westward (eastward) phase lasts a longer time than the eastward (westward) phase below (above) ~60 km. In contrast, the eastward phase lasts a longer time than the eastward phase at 50°N. The standard deviations of the wind differences (left column of Figs. 5d and 5e) vary from ~3 ms$^{-1}$ to ~12 ms$^{-1}$ with the increasing height. The percentage difference is in the range of 10-17% except for ~20% at 38 km and 50°N.

**The comparisons between BU and UarU**

The following descriptions on the wind differences have been added in the text.

[Figure]

**Figure 6.** Latitude-height sections of BU and UraU (the first and third rows) and their differences (the second and fourth rows) in each month (denoted one the left-upper corner of each panel) of a composite year. The BU is represented by color filled contour (positive for eastward, zero wind is highlighted by thick red contour lines). The UraU is represented by contour lines with interval of 10 ms$^{-1}$ (the eastward and westward winds are represented as solid and dash lines, respectively. Zero

wind is highlighted by thick black contour lines). The wind difference is represented by color filled contour and highlighted by contour lines. Same color scale is used in each row.

[revised manuscript text omitted]

**Point 3.** The theoretical basis has been revised as the following (detailed description can be found in response to **Section 2.2**):

Eq. (3) has been successfully applied to the latitude bands of 70°S-8°S and 8°N-70°N to get zonal mean wind (Fleming et al., 1990; Smith et al., 2017). We restrict Eq. (3) at 10°N-50°N and 10°S-50°S due to the un-continuous sampling of the SABER measurements poleward of 53°N/S. At around the equator, the solution of Eq. (3) is an indeterminate form of 0/0 as $\varphi \rightarrow 0$ and can be solved through the L'Hôpital's rule if we get continuous values of $\bar{p}$ and $\bar{\rho}$. In fact, only the discrete values $\bar{p}$ and $\bar{\rho}$ with latitude interval of 2.5° can be obtained from observations. To apply Eq. (3) at the equator, one need to differentiate Eq. (3) with $\varphi$. As $\varphi \rightarrow 0$, we have $\tan\varphi \rightarrow \varphi$, $\sin\varphi \rightarrow \varphi$. Thus, Eq. (3) can be simplified as (Fleming et al., 1990; Swinbank & Ortland, 2003),

$$\bar{u} = -\frac{1}{2\Omega a \bar{\rho}} \frac{\partial^2 \bar{p}}{\partial \varphi^2}. \tag{4}$$

**Comments in detail**

**Section 2.2:** The method of deriving the BU data set is discussed here; this is an important part of the paper. It needs to be clear and should be understandable (in principle) without going back to the cited literature. First Eq. (3) should be valid at the equator (as it can be simplified to Eq. (4) at the equator). Is this correct? But in line 131 you say 10°-50°N/S for the BU data set – this seems to be a contradiction. Further, in l. 136, you say that Eq. (3) is valid from 8°S to 8°N as well as 70°-90°N and 70°-90°S. This is how I read your text. This is inconsistent with the given range of the BU data. I might not be correct here, but this discussion is not as clear as it should be.

**Response:** Thanks for your careful reading. The description should be clarified.

Indeed, Eq. (3) is valid at the equator in theory.

$$\frac{\bar{u}^2}{a}\tan\varphi + f\bar{u} = -\frac{1}{a\bar{\rho}} \frac{\partial \bar{p}}{\partial \varphi}. \tag{3}$$

From Eq. (3), we get

$$\bar{u} = \frac{-f \pm \sqrt{f^2 - 4\frac{\tan\varphi}{a}\frac{1}{a\bar{\rho}}\frac{\partial \bar{p}}{\partial \varphi}}}{2\frac{\tan\varphi}{a}}. \tag{R1}$$

At the equator, $\varphi = 0$. Both the numerator and denominator of the right-hand side (RHS) of Eq. (R1) approaches to 0 as $\varphi \rightarrow 0$. In theory, the RHS of Eq. (R1) is an indeterminate form of 0/0 as $\varphi \rightarrow 0$. In theory, Eq. (R1) can be solved through the L'Hôpital's rule if we get continuous values of $\bar{p}$ and $\bar{\rho}$ as $\varphi \rightarrow 0$. In fact, the continuous values of $\bar{p}$ and $\bar{\rho}$ as $\varphi \rightarrow 0$ cannot be obtained since we have only the discrete values $\bar{p}$ and $\bar{\rho}$ with latitude interval of 2.5°. Thus, although Eq. (3) is valid at the equator, the problem is that we cannot get its solution.

At the equator, one need to differentiate Eq. (3) with $\varphi$. As $\varphi \rightarrow 0$, we have $\tan\varphi \rightarrow \varphi$, $\sin\varphi \rightarrow \varphi$, and $f = 2\Omega\sin\varphi \rightarrow 2\Omega\varphi$, Thus, Eq. (3) can be simplified to

$$\frac{\bar{u}^2}{a} + 2\Omega\bar{u} = -\frac{1}{a\bar{\rho}}\frac{\partial^2 \bar{p}}{\partial \varphi^2} \tag{R2}$$

Since the magnitude of $\bar{u}^2$ (maximum with order of $10^4$) is far less than the radius of the earth $a$ (order of $10^7$), $\bar{u}^2/a \sim 0$. Finally, we get

$$\bar{u} = -\frac{1}{2\Omega a\bar{\rho}}\frac{\partial^2 \bar{p}}{\partial \varphi^2} \tag{4}$$

**In the text, this point has been revised as following:**

Eq. (3) has been successfully applied to the latitude bands of 70°S-8°S and 8°N-70°N to get zonal mean wind (Fleming et al., 1990; Smith et al., 2017). We restrict Eq. (3) at 10°N-50°N and 10°S-50°S due to the un-continuous sampling of the SABER measurements poleward of 53°N/S. At around the equator, the solution of Eq. (3) is an indeterminate form of 0/0 as $\varphi \to 0$ and can be solved through the L'Hôpital's rule if we get continuous values of $\bar{p}$ and $\bar{\rho}$. In fact, only the discrete values $\bar{p}$ and $\bar{\rho}$ with latitude interval of 2.5° can be obtained from observations. To apply Eq. (3) at the equator, one need to differentiate Eq. (3) with $\varphi$. As $\varphi \to 0$, we have $\tan\varphi \to \varphi$, $\sin\varphi \to \varphi$. Thus, Eq. (3) can be simplified as (Fleming et al., 1990; Swinbank & Ortland, 2003),

$$\bar{u} = -\frac{1}{2\Omega a\bar{\rho}}\frac{\partial^2 \bar{p}}{\partial \varphi^2}. \tag{4}$$

**Comparison with reanalyses**

I would be helpful to know if any of the data used for BU are assimilated in MERRA2. Also be clearer about for which latitude range you compare MERRA2 and BU. Why do you not compare with ERA5? (And add the reference to Hersbach et al. (2020) for ERA5).

**Response:** These points should be clarified.

(1) The data used to calculate BU are the temperature and profiles measured by the SABER instrument and the zonal wind observed by a meteor radar at Koto Tabang (0.2°S). None of these data are assimilated in MERRA2. We have clarified this point in the beginning of Section 3.1 as "First of all, we should note that the BU data are derived from the temperature and pressure profiles measured by the SABER instrument and the zonal wind observed by a meteor at Koto Tabang (0.2°S). None of these data are assimilated in assimilated in MERRA2. Thus, BU and MerU are independent.".

(2) In section 2.1, we have refined the latitude range for comparing MERRA2 and BU as "Such that the monthly zonal mean (MerU) wind can be obtained to validate the BU at 50°N-50°S"

(3) Thanks for your references. We have added Hersbach et al. (2020) for ERA5. Moreover, Ern et al. (2021) have performed a comprehensive comparison between ERA5 and MERRA2, as well as some other reanalysis data. Part of results of Ern et al. (2021) have been included in our introduction as "A recent study by Ern et al. (2021) showed that both MERRA2 and ERA5 capture the semiannual oscillations (SAO) in the stratopause region and lower mesosphere at around the equator. In the middle atmosphere, MERRA2 produces a reasonable SAO due to the assimilated Aura Microwave Limb Sounder (MLS) data (Schwartz et al., 2008; Molod et al., 2015). Above 65 km, the mesopause SAO produced by ERA5 is stronger than that by MERRA2. This is because the stronger damping of MERRA2 reduces the amplitude of the mesopause SAO."

ERA5 has spatial resolution of 0.25° in both longitude and latitude and 137 levels (up to ~ 80 km). The main difficulty for us is to get the ERA5 data. Using the CDS API provided on the web of https://confluence.ecmwf.int/display/CKB/ERA5%3A+data+documentation, we have tried to download the ERA5 data. In fact, it takes about several hours to download the data in a model day. These data include the wind (txyz), as well data used to calculate geometric height (e.g., surface pressure (txy), geopotential (txy), temperature (txyz), and relative humidity (txyz)). Here t is hourly time, x=longitude (1440 grids from 180°W to 179.75°E), y=latitude (420 grids from 52.5°S to 52.5°N), z is model level from 1 to 137. Then using these data, the python code "compute_geopotential_on_ml.py" as well as the instructions provided at https://confluence. ecmwf.int/display/CKB/ERA5%3A+compute+pressure+and+geopotential+on+model+levels%2C+ geopotential+height+and+geometric+height, we can get the wind at geometric height. In the future, if we get ERA5 data spanning one year, the comparison will be performed.

    **Response:** We have added a note in the introduction to explain the "tide alias" as "This is because the diurnal tide is prominent and exhibits shorter term (one to serval days) variations. The full diurnal cycle is composed by the data from many days (e.g., 60 days for SABER observations) to obtain diurnal tides, which is the mean of diurnal tides over these days. However, the shorter term variations of diurnal tides are still in the background and alias the derived winds."

---

## Author Response (AR2)

**Responses to Editor:**

**Comments to the author**

Please make final adjustments as suggested by the referee. Please refrain from over-using the term "long-term". Climatology typically uses 31 years as baseline. That should be a benchmark for this word. You could write "a 20-year consistent time series", for example.

**Response:** Thanks for your efforts in evaluating our manuscript. According to your and the reviewer's suggestion, we have replaced the term "long-term" as "18-year internally consistent time series". We use "18-year" but not "20-year" since there are 18 years from 2002 to 2019.

Other minor revisions, such as equation number, symbols and grammar, are also made in this version.

**Responses to Reviewer:**

I see that the authors have invested a lot of effort to improve their manuscript. Many of my concerns are now addressed in the revised version. Many thinks are better explained and outlined in the revised version.

**Response:** Thanks for your careful reading and comments. The point-to-point responses are below.

**I have two remaining comments:**

(1) An argument is made that the data set put forward here is "long-tem": 2002-2019 While 2002-2019 is a long time period, this is much shorter that many reanalyses (e.g. ERA5). I think the point is that one has an internally consistent data set for the period 2002-2019 here.

**Response:** Thanks for your good suggestion. According to your and the editor's suggestion, we have replaced the term "long-term" as "18-year internally consistent time series".

(2) In the derivation of your equations (R2) you compare the value of the radius of the earth with u-bar. But these quantities do not have the same unit so they cannot directly be compared. I think what you want to do is to compare the magnitude of the two terms on the left-hand side of this equation (which have the same unit) and demonstrate that the first term can be neglected against the second term.

**Response:** Thanks for your careful derivations. We use the same equation numbers to revise this point.

Thus, Eq. (3) can be simplified to

$$\frac{\bar{u}^2}{a} + 2\Omega\bar{u} = -\frac{1}{a\bar{\rho}}\frac{\partial^2 \bar{p}}{\partial \varphi^2} \qquad (R2)$$

According to Fleming et al. (1990) and Smith et al. (2017), the monthly mean zonal mean wind is in the range of $\pm75\text{ms}^{-1}$. Thus, the term $\bar{u}^2/a$ is one to two orders smaller than $2\Omega\bar{u}$. After neglecting the term $\bar{u}^2/a$, we can get $\bar{u}$ at the equator as,

$$\bar{u} = -\frac{1}{2\Omega a\bar{\rho}}\frac{\partial^2 \bar{p}}{\partial \varphi^2} \qquad (4)$$

**We have added the following in the text:**

To apply Eq. (3) at the equator, one need to differentiate Eq. (3) with $\varphi$. As $\varphi \to 0$, we have $\tan\varphi \to \varphi$, $\sin\varphi \to \varphi$. Thus, Eq. (3) can be simplified as (Fleming et al., 1990),

$$\frac{\bar{u}^2}{a} + 2\Omega\bar{u} = -\frac{1}{a\bar{\rho}}\frac{\partial^2 \bar{p}}{\partial \varphi^2}. \qquad (4)$$

According to Fleming et al. (1990) and Smith et al. (2017), the monthly mean zonal mean wind is in the range of $\pm75\text{ms}$-1. Thus, the term $\bar{u}^2/a$ is one to two orders smaller than $2\Omega\bar{u}$ and can be neglected. Then, $\bar{u}$ at the equator can be expressed as (Fleming et al., 1990; Swinbank & Ortland, 2003),

$$\bar{u} = -\frac{1}{2\Omega a\bar{\rho}}\frac{\partial^2 \bar{p}}{\partial \varphi^2}. \qquad (5)$$